# *Annotator*: A Generic Active Learning Baseline for LiDAR Semantic Segmentation

**Binhui Xie**
Beijing Institute of Technology
binhuixie@bit.edu.cn

**Shuang Li**✉
Beijing Institute of Technology
shuangli@bit.edu.cn

**Qingju Guo**
Beijing Institute of Technology
qingjuguo@bit.edu.cn

**Chi Harold Liu**
Beijing Institute of Technology
chiliu@bit.edu.cn

**Xinjing Cheng**
Tsinghua University & Inceptio Technology
cnorbot@gmail.com

## Abstract

Active learning, a label-efficient paradigm, empowers models to interactively query an oracle for labeling new data. In the realm of LiDAR semantic segmentation, the challenges stem from the sheer volume of point clouds, rendering annotation labor-intensive and cost-prohibitive. This paper presents *Annotator*, a general and efficient active learning baseline, in which a voxel-centric online selection strategy is tailored to efficiently probe and annotate the salient and exemplar voxel girds within each LiDAR scan, even under distribution shift. Concretely, we first execute an in-depth analysis of several common selection strategies such as Random, Entropy, Margin, and then develop voxel confusion degree (VCD) to exploit the local topology relations and structures of point clouds. *Annotator* excels in diverse settings, with a particular focus on active learning (AL), active source-free domain adaptation (ASFDA), and active domain adaptation (ADA). It consistently delivers exceptional performance across LiDAR semantic segmentation benchmarks, spanning both simulation-to-real and real-to-real scenarios. Surprisingly, *Annotator* exhibits remarkable efficiency, requiring significantly fewer annotations, e.g., just labeling five voxels per scan in the SynLiDAR $\rightarrow$ SemanticKITTI task. This results in impressive performance, achieving 87.8% fully-supervised performance under AL, 88.5% under ASFDA, and 94.4% under ADA. We envision that *Annotator* will offer a simple, general, and efficient solution for label-efficient 3D applications.

## 1 Introduction

3D perception and understanding have become indispensable for machines to effectively interact with the real world. LiDAR (Light Detection And Ranging) [50, 52] is a widely-used methodology for capturing precise geometric information about the environment, spurring significant advancements in areas like autonomous vehicles and robotics [16, 25]. However, semantic segmentation of LiDAR presents an enormous challenge. The high-speed collection of millions of point clouds per second by on-board sensors sharply contrasts with the laborious and cost-prohibitive nature of annotating them. Consider, for instance, the vast number of outdoor scenes an autopilot can encounter, which is practically limitless. Yet, acquiring annotations for these large-scale point clouds entails intensive human labor. This underscores the urgency of establishing a label-efficient learning mechanism capable of boosting performance in the low-data regime [19, 76, 90, 97] or facilitating the adaptation of models to new domains [55, 64, 74, 85].

---

✉ Corresponding author. Project page: `https://binhuixie.github.io/annotator-web/`

37th Conference on Neural Information Processing Systems (NeurIPS 2023).

Extensive solutions encompass semi-supervised [8, 11, 29, 31, 83], weakly-supervised [22, 36, 76, 99] or self-supervised [5, 14, 63, 73, 101] learning. Semi- and weakly-supervised learning methods aim to alleviate the annotation burden by harnessing partially labeled or weakly labeled data. In contrast, self-supervised ones learn representations from point clouds via pretext tasks and then transfer to downstream tasks for weight initialization. Although these works offer scalability and practicality for real-world utility, they also confront new challenges, such as variations in LiDAR configurations, sensor biases, and environmental conditions. That is, the majority of prior works has endeavored to in-distribution scenarios, with limited consideration for label-efficient paradigms in out-of-distribution scenarios, especially for sparse outdoor point clouds. Recent efforts turn to large-scale auxiliary datasets [56, 84] and delve into domain adaptation (DA) algorithms [1, 28, 51, 92] to significantly reduce the annotation workload under a domain

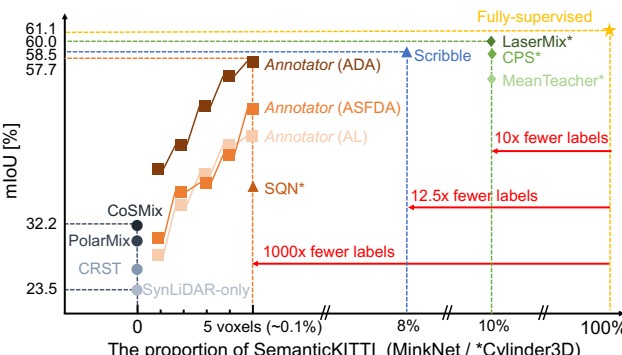

Figure 1: **Performance *vs.* annotated proportion** on SemanticKITTI `val` [3] of existing label-efficient LiDAR segmentation paradigms including domain adaptation (●) [55, 83, 84, 105], weakly- (▲) [22, 76] and semi-supervised (♦) [8, 29, 72] learning. As a reference, fully supervised counterpart (★) is reported as well. *Annotator* (■) attains excellent balance between performance and annotation cost.

shift. Nevertheless, the performance of these methods still lags behind the fully-supervised approaches. In Figure 1, we provide an intuitive comparison of results across various paradigms. It becomes evident that there is ample room for improvement in the performance of these methods.

To surmount these obstacles and promote performance in the domain of interest, active learning (AL) is being an optimal paradigm [27, 35, 60, 82]. Given the limited annotation budget, a common scenario is that only an unlabeled target domain of large amounts of point clouds is available with the goal to interactively select a minimal subset of data to be annotated to maximally improve the segmentation performance. In reality, this setting faces a significant hurdle known as *cold start problem*: the lack of prior information to guide the initial selection of annotated data. A recent work has explored the impact of seeding strategies on the performance of AL methods [58]. Differently, we put forward a new path to access an auxiliary model via pre-training on the open-access auxiliary (source) dataset. This auxiliary model serves as a warm-up stage, allowing for smart target data selection for initial annotation. We formulate this new setting as active source-free domain adaptation, termed ASFDA. Take a further step, drawing inspiration from recent trends in 2D images [38, 44, 87, 88], we delve into the third setting, active domain adaptation (ADA) for semantic segmentation of 3D point clouds. In this setting, a labeled auxiliary dataset is available, and the objective is to select target instances for annotating and learn a model with higher segmentation performance on the target test set.

Overall, in this work, we benchmark three distinct active learning settings for LiDAR semantic segmentation and deliver a simple and general baseline, *Annotator*, as illustrated in Figure 2. Borrowing the idea of modeling and computational techniques in geometry processing, we introduce a voxel-centric selection strategy dedicated to point clouds. Specifically, an input LiDAR scene is first voxelized into voxel grids, with a large voxel size to expand the local areas during the selection process. After obtaining final network predictions, importance estimation is carried out for each voxel grid using several common strategies such as Random, the softmax entropy (Entropy), and the margin between highest softmax scores (Margin). But considering only uncertainty for selection would be suboptimal [2, 44, 86]. Therefore, we introduce the concept of voxel confusion degree (VCD), which takes into account nearby predictions, capturing diversity and redundancy within a voxel grid. VCD enables the exploitation of local topology relations and point cloud structures. As a result, VCD can represent both uncertainty and diversity of a voxel grid in the LiDAR scene. In each active round, we query the top one voxel grid within each scan for annotation until the budget is exhausted. Despite the simplicity of our *Annotator*, it achieves performance on par with the fully-supervised counterpart requiring 1000× fewer annotations and significantly outperforms all prevailing acquisition strategies.

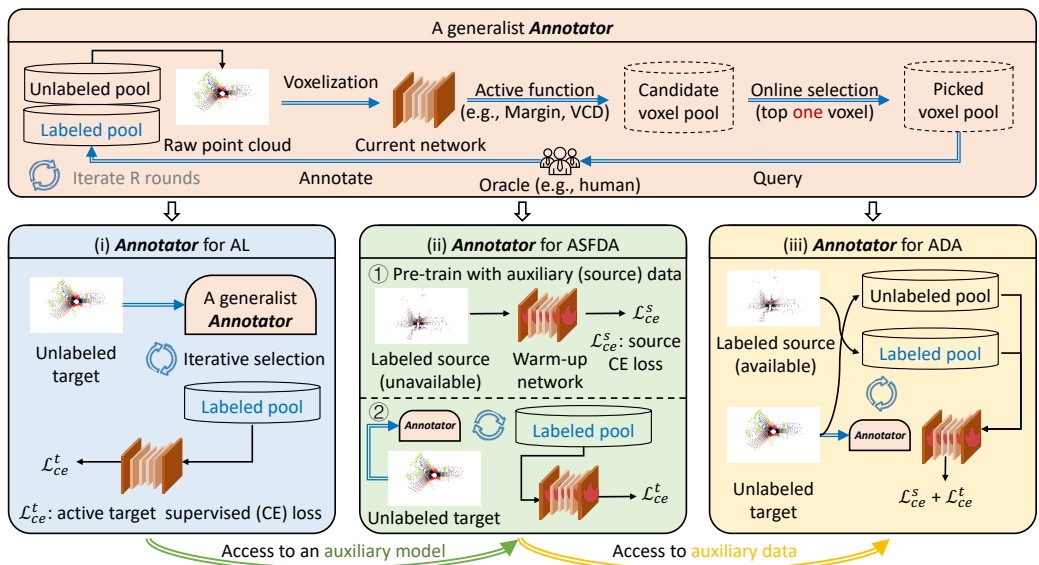

Figure 2: **An illustration of *Annotator***. *Annotator* is a new active learning baseline with broad applicability, capable of interactively querying a tiny subset of the most informative new (target) data points based on available inputs without task-specific designs. This includes (i) only unlabeled new (target) data being available (active learning, AL); (ii) access to an auxiliary (source) pre-trained model (active source-free domain adaptation, ASFDA); and (iii) availability of labeled source data and unlabeled target data (active domain adaptation, ADA). Remarkably, *Annotator* attains excellent results not only in in-domain settings but also manifests adaptive transfer to out-of-domain settings.

The contribution of this paper can be summarized in three aspects. *First*, we present a voxel-centric active learning baseline that significantly reduces the labeling cost and effectively facilitates learning with a limited budget, achieving near performance to that of fully-supervised methods with $1000\times$ fewer annotations. *Second*, we introduce a label acquisition strategy, the voxel confusion degree (VCD), which is more robust and diverse to select point clouds under a domain shift. *Third*, *Annotator* is generally applicable for various network architectures (voxel-, range- and bev-views), settings (in-distribution and out-of-distribution), and scenarios (simulation-to-real and real-to-real) with consistent gains. We hope this work could lay a solid foundation for label-efficient 3D applications.

## 2 A Generic Baseline

### 2.1 Preliminaries and overview

**Problem setup.** In the context of LiDAR semantic segmentation, a LiDAR scan is made of a set of point clouds and let $X \in \mathbb{R}^{N \times 4}$, $Y \in \mathbb{K}^N$ respectively denote $N$ points and the corresponding labels. $\mathbb{K}$ is a predefined semantic class vocabulary $\mathbb{K} = \{1, ..., K\}$ of $K$ categorical labels. Each point $x_i$ in $X$ is a $1 \times 4$ vector with a 3D Cartesian coordinate relative to the scanner $(a_i, b_i, c_i)$ and an intensity value of returning laser beam. Our baseline works in the following settings: active learning (AL), active source-free domain adaptation (ASFDA), and active domain adaptation (ADA). First, we are given an unlabeled target domain $\mathcal{D}^t = \{X^t \cup X^a\}$, where $X^t$ denotes unlabeled target point clouds and $X^a$ denotes the selected points to be annotated and is initialized as empty set, i.e., $X^a = \emptyset$. Next, for ASFDA and ADA, a labeled source domain $\mathcal{D}^s = \{X^s, Y^s\}$ can be utilized only in pre-training stage and anytime respectively. Ultimately, given a limited budget, our goal is to iteratively select a subset of data points from $\mathcal{D}^t$ to annotate until the budget is exhausted, all the while catching up with the performance of the fully-supervised model.

**Overview.** Figure 2 displays an overview of *Annotator*, which is a label-efficient baseline for LiDAR semantic segmentation. It is composed of two parts: 1) a generalist *Annotator* which contains a voxelization process to get voxel grids and an active function with online selection for picking the most valuable voxel grid of each input scan in each active round; 2) the pipelines of distinct active

learning settings are described. For AL, we interactively select a subset of voxels from the current scan to be annotated and train the network with these sparse annotated voxel grids. In the case of ASFDA, we begin by pre-training a network on the source domain through standard supervised learning. This warm-up network then serves as a strong initialization to aid the initial selection. As for ADA, except for the pre-training stage, we also make use of annotated source domain to promote the selection in each round and facilitate domain alignment. In the following, we will detail why we select salient and exemplar data points from a voxel-centric perspective and how to address *cold start problem* via an auxiliary model. After that, overall objectives for all three settings are elaborated.

## 2.2  A generalist *Annotator*

In this section, we proposed a general active learning baseline called *Annotator*. The core idea is to select salient and exemplar voxel grids from each LiDAR scan. It's important to note that previous researches have proposed frame-based [13, 94], region-based [82], and point-based [35] selection strategies. The first two usually require an offline stage, which may be infeasible at large scales. The last one is costly due to the sparsity of outdoor point clouds. By contrast, our voxel-centric selection focuses on querying salient and exemplar areas and annotating all points within those areas. This approach is more efficient and flexible. Moreover, it can be seamlessly applied to various network architectures, including voxel-, range- and bev-views, as demonstrated in the experiment section.

To implement it, we begin with the voxelization process as introduced in [9, 102]. Each input LiDAR scan $X$ is transformed into a 3D voxel grid set $V$. This process involves sampling the continuous 3D input space into discrete voxel grids, where points falling into the same grid are merged. Each voxel grid serves as a selection unit. Mathematically, for a point $x_i \in X$, the corresponding voxel grid coordinate is $(a_i^v, b_i^v, c_i^v) = \lfloor (a_i, b_i, c_i)/\triangle \rfloor$, with $\triangle$ denoting predefined voxel size. In our experiments, we have found that using a large voxel grid is more robust against noise and sparsity. Unless otherwise specified, we use $\triangle_1 = 0.05$ for training and $\triangle_2 = 0.25$ for the selection process.

**Selection strategies.**  For each voxel grid $v_j \in V$, we assess its importance and select the best voxel grid per LiDAR scan in each active round. Initially, we employ a Random selection strategy. Subsequently, we explore softmax entropy (Entropy) and the margin between highest softmax scores (Margin). It's essential to note that while these common selection strategies are not technical contributions, they are necessary to build our baseline. Detailed calculations are provided below.

- **Random**: randomly select a target voxel grid $v_j$ from $V$ to be annotated in each round.
- **Entropy**: first calculate the softmax entropy of each point $x_i \in v_j$ and then adopt the maximum value as the Entropy score of this grid, i.e., $\text{Entropy}(v_j) = \max_{x_i \in v_j} -p_i \log p_i$, where $p_i$ is the softmax score of point $x_i$. The voxel grid with the highest Entropy score is selected in each scan.
- **Margin**: first calculate the margin between highest softmax score of each point $x_i \in v_j$ and then adopt the maximum value as the Margin score of this grid, i.e., $\text{Margin}(v_j) = \max_{x_i \in v_j} (\max(p_i) - \max2(p_i))$, where $\max2(\cdot)$ is the second-largest value operator. In each scan, the voxel grid with the lowest Margin score is chosen.

**The VCD strategy.** Our voxel confusion degree (VCD) is motivated by an important observation: the previously mentioned selection strategies become less effective when models are applied in new domains due to mis-calibrated uncertainty estimation. Therefore, the VCD is designed to estimate category diversity within a voxel grid rather than uncertainty, making it more robust under domain shift. Here's how it works: we begin by obtaining pseudo label $\hat{y}_i$ for each point $x_i$. Next, we divide points within $v_j$ into $K$ clusters: $v_j^{<k>} = \{x_i^{<k>} | x_i \in v_j, \hat{y}_i = k\}$. This allows us to collect statistical information about the categories present in the voxel grid. With this information, we calculate VCD to assess the significance of voxel grids as follows:

$$\text{VCD}(v_j) = -\sum_{k=1}^{K} \frac{|v_j^{<k>}|}{|v_j|} \log \frac{|v_j^{<k>}|}{|v_j|},$$

where $|\cdot|$ denotes the number of points in a set. Finally, voxel grid with the highest VCD score is selected in each scan. The insight is that a higher score indicates a greater category diversity within a voxel, which would be beneficial for model training once being annotated. In all experiments, *Annotator* is equipped with VCD by default, and the results indicate the superiority of VCD strategy.

**Making a good first impression.** To avoid the *cold start problem* mentioned before, we introduce a warm start mechanism that pre-trains an auxiliary model with an auxiliary (source) dataset, and then it is used to select voxel grids in the first round. This warm start stage is applied in ASFDA and ADA.

*Discussion: balancing annotation cost and computation cost.* Our primary focus is on reducing annotation cost while maintaining performance comparable to fully-supervised approaches. Let's consider simulation-to-real tasks as an example. The simplest setup involves active learning within the real dataset. However, this setup yields less satisfactory results due to the *cold-start problem*: the lack of prior information for selecting an initial annotated set. To address this, we utilize a synthetic dataset to train an auxiliary model in a brief warm-up stage, enabling smarter data selection in the first round. Importantly, this warm-up process is short, conducted only once, and results in minimal costs (both annotation and computation). For a detailed analysis, please refer to Appendix B.1.

## 2.3 Optimization

The overall loss function is the standard cross-entropy loss, which is defined as:

$$\mathcal{L}_{ce}(X) = \frac{1}{|X|} \sum_{x_i \in X} \sum_{k=1}^{K} -y_i^k \log p_i^k \,,$$

where $K$ is the number of categories, $y_i$ is the one-hot label of point $x_i$ and $p_i^k$ is the predicted probability of point $x_i$ belonging to category $k$. Hereafter, for AL, the objective is $\min_\theta \mathcal{L}_{ce}(X^a)$; for ASFDA, the objective is $\min_{\theta_s} \mathcal{L}_{ce}(X^a)$; for ADA, the objective is $\min_{\theta_s} \mathcal{L}_{ce}(X^s) + \mathcal{L}_{ce}(X^a)$. Here, $\theta$ and $\theta_s$ denote training from scratch and training from the source pre-trained model, respectively.

# 3 Experiments

In this section, we conduct extensive experiments on several public benchmarks under three active learning scenarios: (i) AL setting where all available data points are from unlabeled target domain; (ii) ASFDA setting where we can only access a pre-trained model from the source domain; (iii) ADA setting where all data points from source domain can be utilized and a portion of unlabeled target data is selected to be annotated. We first introduce the dataset used in this work and experimental setup and then present experimental results of baseline methods and extensive analyses of *Annotator*.

## 3.1 Experiment setup

**Datasets.** We build all benchmarks upon SynLiDAR [84], SemanticKITTI [3], SemanticPOSS [42], and nuScenes [4], constructing two simulation-to-real and two real-to-real adaptation scenarios. SynLiDAR [84] is a large-scale synthetic dataset, which has 198,396 LiDAR scans with point-level segmentation annotations over 32 semantic classes. Following [84], we use 19,840 point clouds as the training data. SemanticKITTI (KITTI) [3] is a popular LiDAR segmentation dataset, including 2,9130 training scans and 6,019 validation scans with 19 categories. SemanticPOSS (POSS) [42] consists of 2,988 real-world scans with point-level annotations over 14 semantic classes. As suggested in [42], we use the sequence 03 for validation and the remaining sequences for training. nuScenes [4] contains 19,130 training scans and 4,071 validation scans with 16 object classes.

**Class mapping.** To ensure compatibility between source and target labels across datasets, we perform class mapping. Specifically, we map SynLiDAR labels into 19 common categories for SynLiDAR → KITTI and 13 classes for SynLiDAR → POSS. Similarly, we map labels into 7 classes for KITTI → nuScenes and nuScenes → KITTI. We refer readers to Appendix A.1 for detailed class mappings.

**Implementation details.** We primarily adopt MinkNet [9] and SPVCNN [71] as the segmentation backbones. Note that, all experiments share the same backbones and are within the same codebase, which are implemented using PyTorch [43] on a single NVIDIA Tesla A100 GPU. We use the SGD optimizer and adopt a cosine learning rate decay schedule with initial learning rate of $0.01$. And the batch size for both source and target data is $16$. For additional details, please consult Appendix A.2. Finally, we evaluate the segmentation performance before and after adaptation, following the typical evaluation protocol [47] in LiDAR domain adaptive semantic segmentation [29, 31, 55, 82, 83].

## 3.2 Experimental results

**Quantitative results summary.** We initially evaluate *Annotator* on four benchmarks and two backbones while adhering to a fixed budget of selecting and annotating five voxel grids in each scan.

| Method | Simulation-to-Real | | Real-to-Real | |
| --- | --- | --- | --- | --- |
| | SynLiDAR $\xrightarrow{19}$ KITTI | SynLiDAR $\xrightarrow{13}$ POSS | KITTI $\xrightarrow{7}$ nuScenes | nuScenes $\xrightarrow{7}$ KITTI |
| Source-/Target-Only | 22.0 / 61.1 | 30.4 / 56.7 | 28.4 / 82.5 | 34.6 / 83.3 |
| Random | 35.3 / 36.3 / 45.3 | 27.4 / 30.9 / 43.4 | 66.0 / 67.5 / 71.9 | 70.9 / 69.7 / 74.7 |
| Entropy [78] | 39.8 / 49.6 / 50.1 | 42.8 / 45.5 / 49.9 | 59.7 / 60.3 / 73.1 | 70.7 / 69.1 / 74.0 |
| Margin [26] | 46.9 / 44.3 / 49.0 | 41.6 / 44.1 / 46.9 | 60.2 / 59.2 / 71.4 | 73.1 / 70.3 / 76.7 |
| *Annotator* | **53.7 / 54.1 / 57.7** | **44.9 / 48.2 / 52.0** | **70.4 / 72.4 / 75.9** | **76.8 / 75.3 / 81.8** |

Table 1: Quantitative summary of all baselines' performance based on MinkNet [9] over various LiDAR semantic segmentation benchmarks using only 5 voxel grids. Source-/Target-Only correspond to the model trained on the annotated source/target dataset which are considered as lower/upper bound. Note that results are reported following the order of AL / ASFDA / ADA in each cell.

| Method | Simulation-to-Real | | Real-to-Real | |
| --- | --- | --- | --- | --- |
| | SynLiDAR $\xrightarrow{19}$ KITTI | SynLiDAR $\xrightarrow{13}$ POSS | KITTI $\xrightarrow{7}$ nuScenes | nuScenes $\xrightarrow{7}$ KITTI |
| Source-/Target-Only | 24.2 / 63.7 | 37.0 / 51.9 | 21.3 / 81.3 | 47.1 / 85.0 |
| Random | 40.9 / 41.7 / 51.0 | 35.5 / 37.8 / 42.3 | 65.0 / 66.9 / 64.3 | 70.4 / 68.1 / 75.8 |
| Entropy [78] | 52.7 / 52.1 / 52.8 | 35.2 / 40.5 / 46.8 | 61.3 / 66.0 / 66.3 | 69.5 / 67.4 / 72.6 |
| Margin [26] | 47.1 / 49.9 / 50.7 | 42.9 / 44.8 / 47.1 | 57.8 / 60.3 / 63.2 | 72.3 / 73.0 / 75.3 |
| *Annotator* | **52.8 / 54.6 / 55.6** | **44.9 / 47.5 / 50.9** | **71.4 / 72.1 / 72.3** | **79.5 / 80.5 / 78.4** |

Table 2: Quantitative summary of all baselines' performance based on SPVCNN [71] over various LiDAR semantic segmentation benchmarks using only 5 voxel grids.

| | Model | car | bi.cle | mt.cle | truck | oth-v. | pers. | b.clst | m.clst | road | park. | sidew. | oth-g. | build. | fence | veget. | trunk | terra. | pole | traff. | mIoU |
| --- | --- | --- | --- | --- | --- | --- | --- | --- | --- | --- | --- | --- | --- | --- | --- | --- | --- | --- | --- | --- | --- |
| | Source-Only | 59.4 | 6.2 | 27.2 | 0.6 | 5.8 | 18.4 | 37.9 | 5.4 | 9.3 | 8.8 | 31.0 | 0.1 | 24.5 | 22.6 | 62.7 | 27.7 | 43.4 | 22.8 | 3.6 | 22.0 |
| DA | ADDA [75] | 52.5 | 4.5 | 11.9 | 0.3 | 3.9 | 9.4 | 27.9 | 0.5 | 52.8 | 4.9 | 27.4 | 0.0 | 61.0 | 17.0 | 57.4 | 34.5 | 42.9 | 23.2 | 4.5 | 23.0 |
| | AdvEnt [77] | 58.3 | 5.1 | 14.3 | 0.3 | 1.8 | 14.3 | 44.5 | 0.5 | 50.4 | 4.3 | 34.8 | 0.0 | 48.3 | 19.7 | 65.5 | 34.8 | 52.0 | 33.0 | 6.1 | 25.8 |
| | CRST [105] | 62.0 | 5.0 | 12.4 | 1.3 | 9.2 | 16.7 | 44.2 | 0.4 | 53.0 | 2.5 | 28.4 | 0.0 | 57.1 | 18.7 | 69.8 | 35.0 | 48.7 | 32.5 | 6.9 | 26.5 |
| | ST-PCT [84] | 70.8 | 7.3 | 13.1 | 1.9 | 8.4 | 12.6 | 44.0 | 0.6 | 56.4 | 4.5 | 31.8 | 0.0 | 66.7 | 23.7 | 73.3 | 34.6 | 48.4 | 39.4 | 11.7 | 28.9 |
| | CoSMix [55] | 75.1 | 6.8 | 29.4 | 27.1 | 11.1 | 22.1 | 25.0 | 24.7 | 79.3 | 14.9 | 46.7 | 0.1 | 53.4 | 13.0 | 67.7 | 31.4 | 32.1 | 37.9 | 13.4 | 32.2 |
| | PolarMix [83] | 76.3 | 8.4 | 17.8 | 3.9 | 6.0 | 26.6 | 40.8 | 15.9 | 70.3 | 0.0 | 44.4 | 0.0 | 68.4 | 14.7 | 69.6 | 38.1 | 37.1 | 40.6 | 10.6 | 31.0 |
| AL | Random | 90.6 | 0.0 | 0.0 | 4.5 | 11.1 | 0.0 | 0.0 | 0.0 | 84.5 | 19.1 | 68.3 | 0.0 | 84.4 | 45.5 | 85.8 | 53.9 | 73.3 | 47.8 | 2.0 | 35.3 |
| | Entropy [78] | 94.2 | 0.0 | 19.8 | 23.4 | 24.7 | 6.4 | 0.0 | 0.2 | 79.0 | 19.5 | 62.4 | 2.4 | 85.1 | 50.4 | 86.9 | 56.5 | 74.2 | 52.9 | 18.6 | 39.8 |
| | Margin [26] | 92.0 | 0.0 | 35.9 | 45.3 | 34.0 | 40.7 | 61.0 | 0.0 | 80.5 | 19.8 | 67.0 | 0.1 | 80.6 | 47.3 | 83.4 | 55.5 | 67.2 | 51.6 | 30.0 | 46.9 |
| | *Annotator* | 94.5 | 0.3 | 40.3 | 56.3 | 46.8 | 63.1 | 76.9 | 0.2 | 84.0 | 23.4 | 69.2 | 2.0 | 87.4 | 51.9 | 85.8 | 62.6 | 70.6 | 61.6 | 43.6 | 53.7 |
| ASFDA | Random | 90.5 | 0.0 | 4.7 | 16.5 | 0.0 | 0.0 | 0.0 | 0.0 | 84.4 | 20.8 | 68.9 | 0.1 | 84.7 | 45.9 | 85.8 | 55.0 | 72.8 | 53.7 | 5.4 | 36.3 |
| | Entropy [78] | 94.1 | 0.0 | 40.7 | 42.6 | 36.1 | 54.1 | 59.9 | 0.3 | 81.1 | 19.3 | 66.3 | 3.3 | 84.6 | 47.8 | 86.3 | 59.6 | 74.4 | 61.4 | 31.0 | 49.6 |
| | Margin [26] | 90.1 | 0.0 | 34.5 | 32.5 | 31.1 | 39.6 | 55.6 | 0.0 | 79.2 | 17.8 | 65.1 | 0.0 | 79.0 | 43.5 | 83.1 | 54.7 | 65.8 | 49.6 | 20.0 | 44.3 |
| | *Annotator* | 94.4 | 0.3 | 34.5 | 78.1 | 47.8 | 59.8 | 60.9 | 1.7 | 84.4 | 21.5 | 70.2 | 3.2 | 87.2 | 54.4 | 86.4 | 65.2 | 73.6 | 60.6 | 44.0 | 54.1 |
| ADA | Random | 93.0 | 0.0 | 30.0 | 23.0 | 25.0 | 37.9 | 32.5 | 0.2 | 84.2 | 25.7 | 71.6 | 0.1 | 81.0 | 54.0 | 83.7 | 56.9 | 72.0 | 53.7 | 35.8 | 45.3 |
| | Entropy [78] | 94.1 | 16.9 | 50.2 | 47.1 | 31.4 | 60.2 | 81.2 | 6.6 | 62.9 | 12.6 | 58.1 | 0.1 | 80.4 | 52.7 | 83.0 | 53.2 | 64.7 | 57.5 | 39.6 | 50.1 |
| | Margin [26] | 92.5 | 0.0 | 39.3 | 58.2 | 30.2 | 51.0 | 76.1 | 0.0 | 87.4 | 22.8 | 68.6 | 0.8 | 69.7 | 51.2 | 77.8 | 55.5 | 61.6 | 57.4 | 36.0 | 49.0 |
| | *Annotator* | 95.2 | 22.0 | 59.7 | 69.0 | 49.4 | 63.4 | 82.1 | 3.6 | 84.1 | 28.9 | 71.4 | 1.7 | 85.4 | 58.8 | 85.6 | 60.1 | 73.2 | 60.3 | 41.6 | 57.7 |
| | Target-Only | 95.7 | 20.4 | 63.9 | 70.3 | 45.5 | 65.0 | 78.5 | 0.0 | 93.5 | 49.6 | 81.0 | 0.2 | 91.1 | 63.8 | 87.2 | 68.5 | 72.3 | 64.4 | 49.1 | 61.1 |

Table 3: Per-class results on task of SynLiDAR $\xrightarrow{19}$ KITTI (MinkNet [9]) using only 5 voxel budgets. Domain adaptation (DA) results are reported from [55, 83].

The results in Table 1 and Table 2 paint a clear picture overall: all baseline methods achieve significant improvements over the Source-Only model, especially for *Annotator* with VCD strategy, underscoring the success of the proposed voxel-centric online selection strategy. In particular, *Annotator* achieves the best results across all simulation-to-real and real-to-real tasks. For SynLiDAR → KITTI task, *Annotator* achieves 87.8% / 88.5% / 94.4% fully-supervised performance under AL / ASFDA / ADA settings respectively. For SynLiDAR → POSS task, they are 79.0% / 85.0% / 91.7% respectively. On the task of KITTI → nuScenes, they are 85.3% / 87.8% / 92.0% respectively. And on the task of nuScenes → KITTI, they are 92.2% / 90.3% / 98.2%, respectively. It is also clear that domain shift between simulation and real-world is more significant than those between real-world datasets. Therefore, simulation-to-real tasks show poorer performance. Further, we compare *Annotator* with additional AL algorithms and extend it to indoor semantic segmentation in Appendix B.2 and B.3.

**Per-class performance.** To sufficiently realize the capacity of our *Annotator*, we also provide the class-wise IoU scores on two simulation-to-real tasks (Table 3 and Table 4) for different algorithms and comparison results with state-of-the-art DA methods [55, 83]. Other results of the remainder

| Model | car | bike | pers. | rider | grou. | buil. | fence | plants | trunk | pole | traf. | garb. | cone. | mIoU |
|---|---|---|---|---|---|---|---|---|---|---|---|---|---|---|
| Source-Only | 44.7 | 1.9 | 33.5 | 38.3 | 77.0 | 54.2 | 30.3 | 63.8 | 22.0 | 12.9 | 0.4 | 11.2 | 4.7 | 30.4 |
| **DA** — CRST [105] | 22.0 | 6.8 | 23.5 | 31.8 | 60.3 | 58.2 | 9.1 | 63.2 | 18.9 | 41.6 | 1.9 | 13.5 | 1.0 | 27.1 |
| ST-PCT [84] | 27.8 | 6.6 | 28.9 | 34.8 | 63.9 | 64.1 | 12.1 | 63.7 | 18.6 | 41.0 | 4.9 | 16.6 | 1.6 | 29.6 |
| CoSMix [55] | 36.2 | 10.6 | 55.8 | 51.4 | 78.7 | 66.2 | 24.9 | 71.3 | 23.5 | 34.2 | 22.5 | 28.9 | 20.4 | 40.4 |
| PolarMix [83] | 25.0 | 10.7 | 32.6 | 39.1 | 79.0 | 44.8 | 23.8 | 64.2 | 11.9 | 29.6 | 5.8 | 15.3 | 13.3 | 30.4 |
| **AL** — Random | 24.0 | 47.8 | 28.9 | 0.1 | **79.3** | **66.7** | 27.7 | **76.4** | 0.1 | 5.5 | 0.2 | 0.0 | 0.0 | 27.4 |
| Entropy [78] | 37.8 | 39.6 | **58.9** | 45.2 | 75.2 | 56.3 | 38.6 | 69.7 | **39.3** | 23.7 | **36.1** | 1.6 | **34.2** | 42.8 |
| Margin [26] | 30.1 | 44.2 | 55.3 | 46.8 | 79.2 | 63.7 | 44.0 | 74.3 | 34.1 | 23.4 | 34.5 | 9.7 | 1.2 | 41.6 |
| *Annotator* | **41.0** | **50.1** | 49.3 | **52.0** | 78.5 | 66.4 | **56.4** | 73.4 | 31.1 | **29.6** | 34.5 | **15.7** | 6.2 | **44.9** |
| **ASFDA** — Random | 32.2 | 46.4 | 37.9 | 0.4 | 79.2 | **69.8** | 33.0 | **77.7** | 17.4 | 5.5 | 2.8 | 0.0 | 0.0 | 30.9 |
| Entropy [78] | 32.1 | 46.5 | **65.7** | 58.0 | 74.4 | 62.9 | 45.5 | 69.6 | **41.5** | **34.5** | 33.7 | 13.2 | **14.3** | 45.5 |
| Margin [26] | 30.2 | 47.6 | 59.5 | 44.5 | 79.7 | 66.8 | 51.7 | 73.5 | 28.6 | 30.1 | **35.2** | **25.2** | 0.1 | 44.1 |
| *Annotator* | **56.2** | **54.2** | 63.6 | **58.7** | **80.9** | 64.6 | **58.1** | 73.4 | 37.8 | 26.3 | 34.0 | 6.3 | 11.9 | **48.2** |
| **ADA** — Random | 65.0 | 10.9 | 59.3 | 54.3 | 58.6 | 70.0 | **54.2** | 63.9 | 39.6 | **39.8** | 20.8 | 27.8 | 0.0 | 43.4 |
| Entropy [78] | 53.3 | **29.1** | 62.9 | 52.7 | **80.3** | **71.9** | 48.2 | **72.3** | 38.9 | 30.0 | 27.6 | 44.2 | 37.8 | 49.9 |
| Margin [26] | 61.3 | 25.2 | 60.6 | **56.2** | 79.6 | 54.2 | 46.6 | 66.7 | 38.1 | 29.2 | 30.8 | 40.8 | 20.4 | 46.9 |
| *Annotator* | **67.4** | 18.0 | **64.0** | 52.0 | 78.5 | 61.5 | 1.5 | 68.6 | **48.5** | 32.7 | **37.9** | **50.8** | **43.8** | **52.0** |
| Target-Only | 73.7 | 60.4 | 68.6 | 62.2 | 81.7 | 79.2 | 60.8 | 78.9 | 36.5 | 31.2 | 44.1 | 12.9 | 46.6 | 56.7 |

Table 4: Per-class results on task of SynLiDAR $\xrightarrow{13}$ POSS (MinkNet [9]) using only 5 voxel budgets.

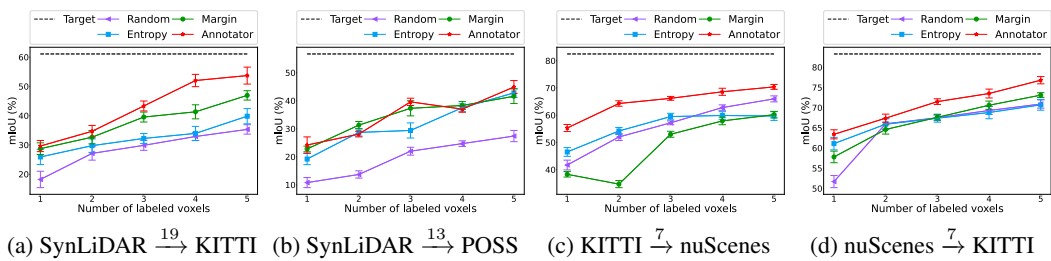

(a) SynLiDAR $\xrightarrow{19}$ KITTI  (b) SynLiDAR $\xrightarrow{13}$ POSS  (c) KITTI $\xrightarrow{7}$ nuScenes  (d) nuScenes $\xrightarrow{7}$ KITTI

Figure 3: Active learning results on various benchmarks varying active budget.

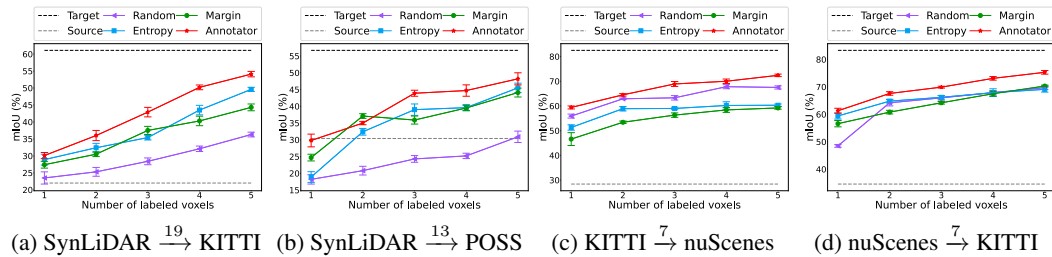

(a) SynLiDAR $\xrightarrow{19}$ KITTI  (b) SynLiDAR $\xrightarrow{13}$ POSS  (c) KITTI $\xrightarrow{7}$ nuScenes  (d) nuScenes $\xrightarrow{7}$ KITTI

Figure 4: Active source-free domain adaptation results on various benchmarks varying active budget.

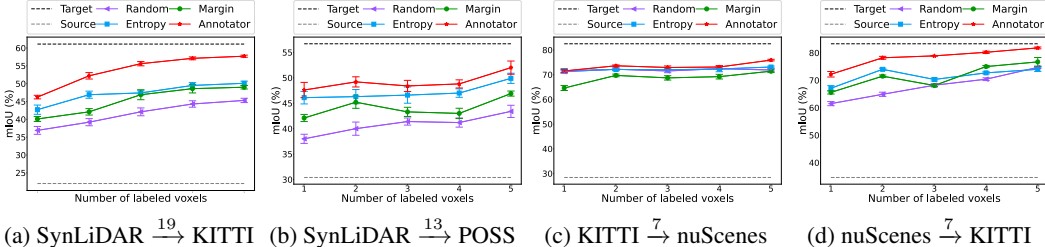

(a) SynLiDAR $\xrightarrow{19}$ KITTI  (b) SynLiDAR $\xrightarrow{13}$ POSS  (c) KITTI $\xrightarrow{7}$ nuScenes  (d) nuScenes $\xrightarrow{7}$ KITTI

Figure 5: Active domain adaptation results on various benchmarks varying active budget.

tasks and backbones are listed in Appendix B.4. It is noteworthy that *Annotator* under any active learning settings significantly outperform DA methods with respect to some specific categories such as "traf.", "pole", "garb." and "cone" etc. These results also showcase the class-balanced selection of the proposed *Annotator*, which is testified in Figure 6 as well.

**Results with varying budgets.**  We investigate the impact of varying budgets and compare the performance with baseline methods, as illustrated in Figure 3, Figure 4 and Figure 5. A consistent

| | Model | car | bike | pers. | rider | grou. | buil. | fence | plants | trunk | pole | traf. | garb. | cone. | mIoU |
|---|---|---|---|---|---|---|---|---|---|---|---|---|---|---|---|
| SalsaNet | Random | 30.9 | 40.6 | 22.8 | 10.4 | 74.7 | **57.2** | 26.6 | **66.6** | 15.6 | 5.5 | 8.3 | 0.0 | 10.6 | 28.4 |
| | Entropy | **32.8** | **45.2** | 33.1 | 18.6 | **76.8** | 52.6 | 40.2 | 64.7 | 20.1 | 5.5 | 11.2 | 12.7 | 4.3 | 32.1 |
| | Margin | 29.8 | 38.2 | 33.6 | 28.0 | 71.1 | 48.0 | 27.7 | 61.3 | 24.5 | **12.5** | **20.6** | **18.4** | 0.0 | 31.8 |
| | **Annotator** | 32.0 | 45.1 | **39.7** | **31.8** | 76.5 | 53.9 | **40.9** | 64.7 | **26.2** | 11.8 | 17.7 | 13.7 | **13.1** | **35.9** |
| | Target-Only | 39.2 | 51.0 | 52.7 | 40.2 | 79.3 | 66.1 | 50.1 | 71.5 | 28.1 | 18.7 | 28.3 | 8.0 | 16.7 | 42.3 |
| PolarNet | Random | 36.6 | 50.5 | 40.8 | 0.1 | 76.1 | 50.3 | 50.3 | **74.0** | 3.1 | 17.8 | 1.3 | 0.0 | 0.0 | 32.3 |
| | Entropy | **44.5** | 48.8 | 50.3 | 11.8 | **77.9** | 63.6 | 45.4 | 71.0 | 10.0 | 13.3 | 19.1 | 0.0 | 0.0 | 35.0 |
| | Margin | 22.0 | 35.4 | 42.8 | 24.3 | 64.1 | 54.7 | 33.0 | 64.2 | 19.4 | **20.1** | 17.5 | 4.0 | 0.0 | 30.9 |
| | **Annotator** | 44.4 | **51.7** | **55.9** | **39.2** | 76.2 | 64.3 | **51.9** | 70.3 | **22.4** | 18.6 | **28.7** | 6.9 | **21.7** | **42.5** |
| | Target-Only | 66.3 | 57.2 | 62.3 | 51.8 | 80.8 | 74.9 | 61.3 | 75.5 | 22.8 | 21.8 | 29.4 | 4.8 | 46.1 | 50.4 |

Table 5: Per-class results on the SemanticPOSS `val` (range-view: SalsaNet [10] and bev-view: PolarNet [100]) under active learning setting using only 10 voxel budgets.

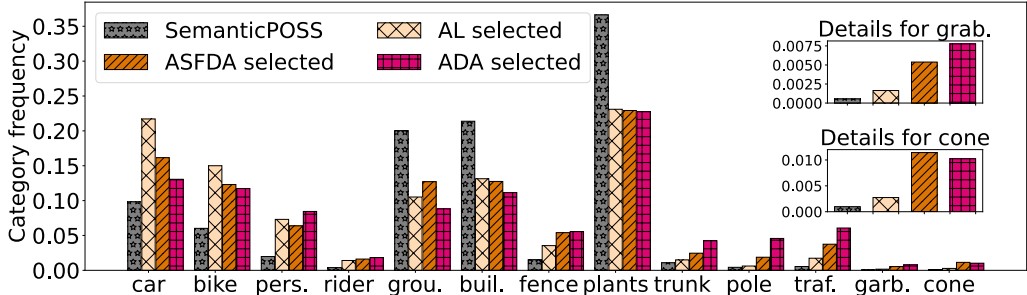

Figure 6: **Category frequencies** on SemanticPOSS `train` [42] of *Annotator* selected 5 voxel grids under AL, ASFDA, ADA scenarios, with the model trained on SynLiDAR $\xrightarrow{13}$ POSS (MinkNet [9]).

observation across these experiments is that *Annotator* consistently outperforms the baseline methods regardless of the budget allocation. In particular, *Annotator* achieves the best performance with about five voxel grids of each LiDAR scan, highlighting the effectiveness of our method in selecting informative areas for active learning. Additionally, we notice that the performance of *Annotator* tends to saturate when the budget exceeds four voxel grids, particularly in the nuScenes → KITTI adaptation task. This phenomenon can be attributed to the fact that the selected voxels at this point provide a sufficient foundation for training a highly competent segmentation model.

## 3.3 Analysis

**More network architectures.** As a general baseline, *Annotator* can be easily applied to other non-voxelization based backbones. Here, we conduct experiments on both SalsaNext [10] (range-view) and PolarNet [100] (bev-view) and per-class results are presented in Table 5. The findings reveal that *Annotator* continues to yield significant gains, even when applied to range- or bev-view backbones, with a limited budget. However, the performance gains in these cases are somewhat less pronounced compared to the voxel-view counterparts. Also, to achieve a fully-supervised performance of 85%, a budget twice as large is required. We suspect that some annotations derived from voxel-centric selection may not be entirely applicable to other non-voxelization based methods.

**Effect of voxel size $\triangle_2$.** We conduct experiments on different $\triangle_2$ while keeping the same budget for selection process and results are listed in Table 6. We can observe that active rounds (# round) decreases as $\triangle_2$ increases since the number of voxels (# voxel) will be small when $\triangle_2$ is large. Notably, the performance of the large voxel grid ($\geq 0.2$) is more adequate and robust.

| $\triangle$ | 0.05 | 0.1 | 0.15 | 0.2 | 0.25 | 0.3 | 0.35 |
|---|---|---|---|---|---|---|---|
| # voxel | 64973 | 54543 | 43795 | 36091 | 30414 | 25992 | 22539 |
| # round | 11 | 9 | 7 | 6 | 5 | 4 | 4 |
| AL | 39.6 | 42.9 | 43.9 | 44.2 | 44.9 | 45.1 | 44.8 |
| ASFDA | 40.0 | 46.2 | 46.0 | 48.0 | 48.2 | 48.3 | 48.0 |
| ADA | 44.5 | 44.1 | 49.3 | 52.4 | 52.0 | 52.1 | 51.4 |

Table 6: Experiments on different values of $\triangle_2$ (from 0.05 to 0.35) for selection process, conducted on SynLiDAR $\xrightarrow{13}$ POSS (MinkNet [9]).

**Category frequency.** To obtain deep insight into *Annotator*, we also visualize a detailed plot of the class frequencies of 5 voxel grids selected by *Annotator* (AL, ASFDA. ADA) and true class

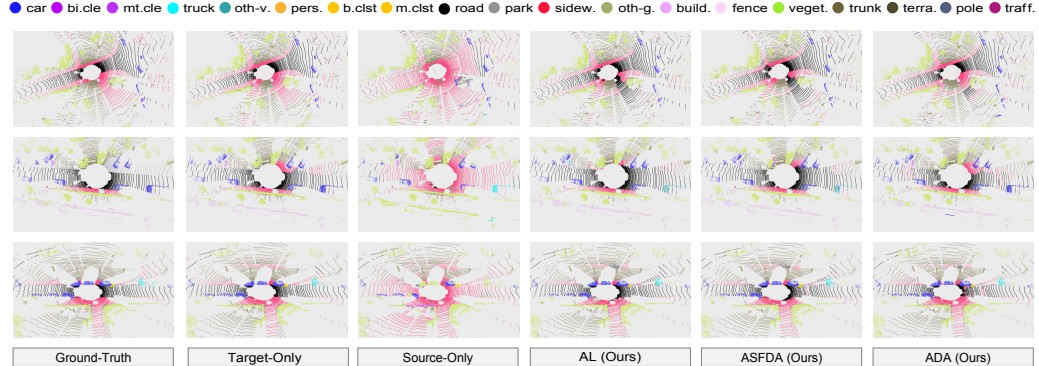

● car ● bi.cle ● mt.cle ● truck ● oth-v. ● pers. ● b.clst ● m.clst ● road ● park ● sidew. ● oth-g. ● build. ● fence ● veget. ● trunk ● terra. ● pole ● traff.

| Ground-Truth | Target-Only | Source-Only | AL (Ours) | ASFDA (Ours) | ADA (Ours) |

Figure 7: **Visualization of segmentation results** for the task SynLiDAR $\xrightarrow{19}$ KITTI using MinkNet [9]. Each row shows results of Ground-Truth, Target-Only, Source-Only, our *Annotator* under AL, ASFDA, and ADA scenarios one by one. Best viewed in color.

frequencies of the SemanticPOSS `train` in Figure 6. As expected, we clearly see that the true distribution is exactly a long-tail distribution while *Annotator* is able to pick out more voxels that contain rare classes. Particularly, it asks labels for more annotations of "rider", "pole", "trunk", "traf.", "grab." and "cone". This, along with the apparent gains in these classes in Table 4, confirms the diversity and balance of voxel grids selected by *Annotator*.

**Qualitative results.** Figure 7 visualizes segmentation results for Source-/Target-Only, our *Annotator* under AL, ASFDA, and ADA approaches on SemanticKITTI `val`. The illustration demonstrates the ability of *Annotator* to enhance predictions not only in distant regions but also to effectively eliminate false predictions across a wide range of directions around the center. By employing voxel-centric selection, *Annotator* successes in enhancing segmentation accuracy even when faced with extremely limited annotation availability. More qualitative results are shown in Appendix B.5.

## 3.4 Limitations

Currently, *Annotator* has two main limitations. First, high annotation cost and potential biases. *Annotator* has made substantial strides in enhancing LiDAR semantic segmentation with human involvement. Nonetheless, the annotation cost remains a challenge. It's imperative to acknowledge the existence of label and sensor biases, which can be a safety concern in real-world deployments. Second, expansion beyond semantic segmentation. *Annotator* current focus on LiDAR semantic segmentation represents a significant limitation in fully realizing its potential. In the future work, we plan to extend *Annotator* to other 3D tasks, such as LiDAR object detection. This may involve two key changes: i) shifting to frame-level selection; ii) reformulating the VCD strategy to consider the diversity for each box annotation.

## 4 Related Work

**LiDAR perception.** Deep learning has made LiDAR perception tasks such as classification [17, 49, 70, 98] and detection [7, 30, 65, 93] easy to solve, allowing deployment in outdoor scenarios. Differently, LiDAR semantic segmentation [3, 23, 32, 33, 40, 45, 46], receiving a class label for each point, is an indispensable technology to understand a scene that is beyond the scope of modern object detectors [35]. There exist various techniques to segment the 3D LiDAR point clouds, e.g., point [32, 45, 79], voxel [18, 39, 103], range [80, 81], bird's eye [100], and multiple view [68, 89, 96] methods. As the best approaches for LiDAR perception are typically trained under full supervision, which can be costly more than capturing data itself, several methods resort to more frugal learning techniques [14], such as semi- [29, 31], weak- [22, 34] and self-supervision [73, 101], zero-shot [21] and few-shot [63] learning and, as studied here, active learning [66] and domain adaptation [74].

**Active learning for LiDAR point clouds.** To avoid the burden of complete point cloud annotation, these methods iteratively select and request the most exemplar scans [94], regions [82], points [35], or

boxes [37] to be labeled during the network training. Most selection strategies lean on uncertainty [24, 88] or diversity [60, 91] criteria. Uncertainty sampling can be measured over each point prediction scores of the model, e.g., softmax entropy [78] or the margin between the two highest scores [53], to select the most confusing of the current model. For example, Hu *et al.* [24] estimate the inconsistency across frames to exploit the inter-frame uncertainty embedded in LiDAR sequences. On the other side, diversity sampling has been ensured by selecting core sets [59]. Leveraging the unique geometric structure of LiDAR point clouds, Liu *et al.* [35] partition the point could into a collection of components then annotate a few points for each component. Recently, the need for an initially annotated fraction of the data to bootstrap an active learning method has been investigated [6, 20, 58, 95, 104], which is termed as cold start problem. In this work, we show that a smart selection of the first set of data with the aid of an auxiliary model can boost all baseline methods drastically.

**Domain adaptation for LiDAR point clouds.**   To tackle the sensor-bias problem encountered in LiDAR deployment, a large body of literature on domain adaptation (DA) [12, 28, 54–57, 74, 92] has been developed. These methods aim to overcome the challenges posed by variations in data collection, sensor characteristics, and environmental conditions, enabling machines to perceive the real world more accurately and reliably. To name a few, Kong *et al.* [28] explore cross-city adaptation for uni-modal LiDAR segmentation. Rochan *et al.* [51] propose a self-supervised adaptation technique with gated adapters. Saltori *et al.* [55] mitigate the domain shift by creating two new intermediate domains via sample mixing. Similarly, with the intermediate domain, Ding *et al.* [12] propose a data-oriented framework with a pretraining and a self-training stage for 3D indoor scenes. Despite the significant progress made in DA, the label scarcity of target domain severely handicaps its utility as the performance of such models often lags far behind the supervised learning counterparts. With this consideration, given an acceptable annotation budget, we explore a simple annotating strategy to assist adaptation process and significantly boost the performance of target domain.

Up to now, active learning coupled with domain adaptation has great practical significance [15, 38, 41, 44, 48, 62, 61, 67, 69, 86]. Nevertheless, rather little work has been done to consider the problem in 3D domains. A recent effort, UniDA3D [13], effectively tackles domain adaptation and active domain adaptation tasks for 3D semantic segmentation. UniDA3D employs a unified multi-modal sampling strategy, selecting informative pairs of 2D-3D data from both source and target domains through a domain discriminator, primarily for ADA tasks. The primary distinction is that our *Annotator* serves as a benchmark for active learning, active source-free domain adaptation, and active domain adaptation tasks, delivering a simple and general AL algorithm for LiDAR point clouds. *Annotator* focuses on enabling AL in both in-distribution and out-of-distribution scenarios. In contrast, UniDA3D places a greater emphasis on adaptation tasks. On the other hand, *Annotator* minimizes human labor in a new domain, regardless of the availability of samples from an auxiliary domain. Methodically, *Annotator* adopts a voxel-centric representation for structured LiDAR data, which is different from the scan-based representation in UniDA3D. Furthermore, *Annotator* is more efficient than UniDA3D in terms of both computation and annotation cost.

## 5   Conclusion

In this work, we present *Annotator*, a generalist active learning baseline, to tackle LiDAR semantic segmentation under three distinct label-efficient settings: active learning (AL), active source-free domain adaptation (ASFDA), and active domain adaptation (ADA). *Annotator* harnesses the power of a purpose-designed voxel confusion degree selection strategy, enabling it to make optimal use of limited budgets while achieving efficient selection and effective performance. Experiments conducted on widely-used simulation-to-real and real-to-real LiDAR semantic segmentation benchmarks demonstrate a substantial performance improvement. Looking forward, we believe the effectiveness and simplicity of *Annotator* has the potential to serve as a powerful tool for label-efficient 3D applications.

## Acknowledgements

This paper was supported by National Key R&D Program of China (No. 2021YFB3301503), the National Natural Science Foundation of China (No. 62376026), and also sponsored by Beijing Nova Program (No. 20230484296).

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
