# Supplemental Material - *Annotator*: A Generic Active Learning Baseline for LiDAR Semantic Segmentation

**Binhui Xie**
Beijing Institute of Technology
binhuixie@bit.edu.cn

**Shuang Li**[✉]
Beijing Institute of Technology
shuangli@bit.edu.cn

**Qingju Guo**
Beijing Institute of Technology
qingjuguo@bit.edu.cn

**Chi Harold Liu**
Beijing Institute of Technology
chiliu@bit.edu.cn

**Xinjing Cheng**
Tsinghua University & Inceptio Technology
cnorbot@gmail.com

## A    Implementation details

### A.1    Dataset

**SynLiDAR [25]** is a large-scale synthetic dataset that is captured with the Unreal Engine [6]. It has 13 LiDAR point cloud sequences with 198,396 scans in total, where each scan has around 98,000 points on average. Precise point-wise annotations of 32 semantic classes are provided for fine-grained 3D scene understanding. It includes 12 LiDAR point cloud sequences (sequence 00 to 11) and has 19,840 point clouds for training following the authors' instructions [25].

**SemanticKITTI [2]** is a comprehensive autonomous driving dataset consisting of LiDAR acquisitions of famous KITTI Vision Odometry Benchmark [7, 8]. The LiDAR point clouds are captured in Karlsruhe (Germany) by a 64-beam LiDAR sensor, with point-level annotations over 19 semantic classes. It includes 22 LiDAR point cloud sequences that are split into a `train` set (sequence 00 to 10, where 08 is used for validation) and a `test` set (sequence 11 to 21). Following [17, 18, 24, 25], we do not use the `test` set, and only use the `train` set for training and validation in all experiments.

**SemanticPOSS [15]** consists of 2,988 real-world scans with point-level annotations over 14 semantic classes. The data is collected in Peking University and uses the same data format as SemanticKITTI. It includes 6 LiDAR point cloud sequences (sequence 00 to 05) and we use the sequence 03 for validation and the remaining sequences for training based on the official benchmark guidelines [15].

**nuScenes [3]** is another large-scale LiDAR segmentation dataset widely adopted in academia. It provides 1,000 driving scenes, where each scene is collected by a 32-beam LiDAR sensor from Boston and Singapore. We follow the official `train` and `val` sample splittings. The total number of LiDAR scans is 40000. The training and validation sets contain 28130 and 6019 scans, respectively

**Class mapping.**    To ensure all tasks are well-defined, we formalize consistent and compatible semantic class vocabulary across the above datasets, ensuring there is a one-to-one mapping between all semantic classes. Table A1 summarizes the unified label space for SynLiDAR [25], SemanticKITTI [2], SemanticPOSS [15], and nuScenes [3].

### A.2    Training details

**Model configuration.**    For our main experiments, we employ two common network architectures: MinkNet [4] and SPVCNN [20]. The voxel size $\triangle_1 = 0.05$ for training and we adopt coordinates and intensity of point clouds as input features. For non-voxelization backbones, we set the range

---

[✉] Corresponding author.

37th Conference on Neural Information Processing Systems (NeurIPS 2023).

| SynLiDAR | SemanticKITTI | SemanticPOSS | nuScenes | $\xrightarrow{19}$ | $\xrightarrow{13}$ | $\xrightarrow{7}$ |
|---|---|---|---|---|---|---|
| car | car
moving-car | car | vehicle.car | 1 | 1 | 1 |
| bicycle | bicycle | bike | vehicle.bicycle | 2 | 2 | |
| truck | truck
moving-truck | | | 4 | | |
| motorcycle | motorcycle | | | 3 | | |
| bus | bus
moving-bus | | | 5 | | |
| sidewalk | sidewalk | | flat.sidewalk | 11 | | 4 |
| female
male
kid | person
moving-person | 1 person
2+ person | human.pedestrian.adult
human.pedestrian.police_officer
human.pedestrian.child
human.pedestrian.construction_worker | 6 | 3 | 2 |
| vegetation | vegetation | plants | | 15 | 8 | |
| road | road
lane-marking | ground | flat.driveable_surface | 9 | 5 | 3 |
| terrain | terrain | | flat.terrain | 17 | | 5 |
| other-ground | other-ground | | flat.other | 12 | | |
| pole | pole | pole | | 18 | 10 | 6 |
| other-vehicle | other-vehicle
on-rails
moving-on-rails
moving-other | | | 5 | | |
| building | building | building | | 13 | 6 | 6 |
| bicyclist | bicyclist
moving-bicyclist | | | 7 | 4 | 4 |
| trunk | trunk | trunk | | 16 | 9 | 7 |
| traffic-sign | traffic-sign | traffic sign 1
traffic sign 2
traffic sign 3 | | 19 | 11 | 6 |
| parking | parking | | | 10 | | 3 |
| motorcyclist | motorcyclist
moving-motorcyclist | | | 8 | 4 | |
| fence | fence | fence | | 14 | 7 | 6 |
| garbage-can | | garbage-can | | | 12 | |
| traffic-cone | | cone/stone | movable.trafficcone | | 13 | |
| | | rider | | | 4 | |
| | | | static.manmade | | | 6 |

Table A1: Unified label space for SynLiDAR, SemanticKITTI, SemanticPOSS, nuScenes: there are over 50 object categories and we list them for individual datasets. In details, we also list training IDs for SynLiDAR $\xrightarrow{19}$ KITTI, SynLiDAR $\xrightarrow{13}$ POSS, KITTI $\xrightarrow{7}$ nuScenes, and nuScenes $\xrightarrow{7}$ KITTI.

image size to 1024×64 for SalsaNext [5] (range-view). We extract point features and set the grid size to (480, 360, 32) for PolarNet [26] (bev-view). All these networks start from randomly initialized weights. As for ASFDA and ADA settings, we have an additional warm-up stage, i.e., the network is pre-trained on the corresponding source domain for 10 epochs with the standard cross-entropy loss.

**Training configuration.** All methods are implemented using PyTorch [16] on a single NVIDIA Tesla A100 GPU. We utilize the SGD optimizer with an initial learning rate of $0.01$. The training process spans 50 epochs and a cosine learning rate decay schedule is also applied for stable training. Both source and target data have a batch size of 16. For our voxel-centric active learning baseline, we maintain $\triangle_2 = 0.25$ for the selection process, unless otherwise specified.

# B  Additional experimental results

## B.1  Computation cost and annotation cost

As previously discussed in the method section, striking a balance between computation cost and annotation cost is a crucial challenge in active learning. In Table A2, we provide a comprehensive breakdown of the computation cost for the SynLiDAR → KITTI task. This highlights the ability of *Annotator* to achieve an optimal equilibrium between high performance and low cost, encompassing both computation and annotation expenses. In the future, we are committed to exploring even more efficient strategies to further reduce the costs associated with both computation and annotation.

| phase | total epoch | running time (hour) | mIoU |
|---|---|---|---|
| pre-train on SynLiDAR | 10 | 2.34 | 22.0 |
| active learning | 50 | 18.04 | 53.7 |
| active source-free domain adaptation | 50 | 18.39 | 54.1 |
| active domain adaptation | 50 | 28.48 | 57.7 |

Table A2: Computation cost analysis for the SynLiDAR → KITTI task.

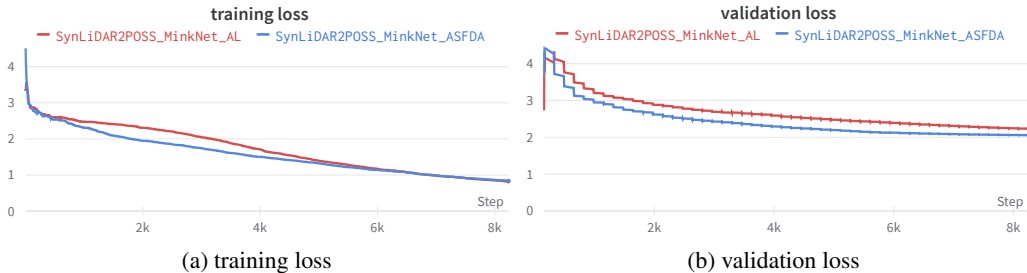

(a) training loss

(b) validation loss

Figure A1: Training and validation loss curves on the task of SynLiDAR → POSS under AL and ASFDA settings (MinkNet [4]).

## B.2 Training curves

In Figure A1, we present the training and validation loss curves for the SynLiDAR → POSS task under both AL and ASFDA settings. Both training loss and validation loss consistently decrease over time, indicating effective model training. Notably, the final validation loss is smaller than the training loss, suggesting a lack of overfitting. Another interesting observation is that the validation loss of the ASFDA approach is smaller than that of the AL approach, underscoring the potency of the auxiliary model in enhancing model performance.

## B.3 Comparison with existing active learning methods

We report mIoU results across existing AL approaches in Table A3. Notably, while LESS [13] obtains the best results with the fewest point labels, it does so by incorporating a complex pre-segmentation stage. In contrast, *Annotator* with a simpler baseline manages to deliver promising results.

| Method | Budget | MinkNet | SPVCNN | Cylinder3D |
|---|---|---|---|---|
| ReDAL [23] | 1% | 47.5 | 48.5 | - |
| LiDAL [10] | 1% | 37.8 | 42.6 | |
| LESS [13] | 0.01% | - | - | 61.0 |
| ***Annotator*** | 0.1% | 53.7 | 52.8 | - |

Table A3: Performance comparison on the SemanticKITTI `val` under active learning setting.

| Method | Random | Entropy | Margin | SSDR-AL | ***Annotator*** |
|---|---|---|---|---|---|
| Total budget | 40.9% | 46.7% | 43.0% | 11.7% | 9.9% |

Table A4: Comparing the percentage of labeled points required to achieve 90% accuracy on S3DIS dataset for different active learning methods.

## B.4 Comparison with indoor semantic segmentation methods

Following SSDR-AL [19], we apply *Annotator* to indoor semantic segmentation task and conduct experiments on the S3DIS [1] dataset. In Table A4, we compare the percentage of labeled points required to achieve 90% accuracy across various methods based on RandLA-Net [9]. It's noteworthy

that *Annotator* is able to annotate 1.8% fewer points than SSDR-AL in achieving the 90% performance of the fully-supervised method.

## B.5 Per-class performance

Table A5 and Table A6 provide the class-wise IoU scores on two real-to-real tasks using MinkNet [4] for different algorithms and comparison results with state-of-the-art DA methods [12, 14, 17, 18, 22].

Table A7 - A10 provide the detailed class-wise IoU scores based on SPVCNN [20].

| | Model | vehicle | person | road | sidewalk | terrain | manmade | vegetation | mIoU |
|---|---|---|---|---|---|---|---|---|---|
| | Source-Only | 47.1 | 1.6 | 52.6 | 14.6 | 2.0 | 33.3 | 47.9 | 28.4 |
| DA | Mix3D [14] | 33.7 | 11.2 | 58.5 | 12.9 | 5.3 | 50.4 | 48.6 | 31.5 |
| | CoSMix [17] | 35.9 | 0.0 | 58.1 | 11.6 | 9.0 | 45.2 | 49.1 | 29.8 |
| | SN [22] | 21.4 | 0.0 | 60.5 | 15.1 | 6.2 | 31.9 | 45.7 | 25.8 |
| | RayCast [12] | 28.8 | 0.0 | 59.3 | 16.1 | 12.5 | 49.7 | 49.8 | 30.9 |
| | LiDOG [18] | 24.0 | 14.9 | 70.6 | 24.6 | 14.0 | 45.3 | 50.9 | 34.9 |
| AL | Random | 83.4 | 15.7 | 90.7 | 48.5 | 65.0 | **81.2** | **77.6** | 66.0 |
| | Entropy [21] | 86.2 | 0.0 | 88.1 | 38.1 | 64.8 | 72.8 | 67.8 | 59.7 |
| | Margin [11] | 82.6 | 0.0 | 86.3 | 38.0 | 60.4 | 78.9 | 75.0 | 60.2 |
| | *Annotator* | **88.1** | **44.2** | **91.9** | **56.7** | **67.1** | 75.5 | 69.5 | **70.4** |
| ASFDA | Random | 85.0 | 23.7 | 89.9 | 48.6 | 65.3 | **81.6** | **78.0** | 67.5 |
| | Entropy [21] | 86.3 | 0.0 | 88.3 | 42.8 | 64.3 | 73.9 | 66.3 | 60.3 |
| | Margin [11] | 81.7 | 0.0 | 86.1 | 39.0 | 58.1 | 77.0 | 72.4 | 59.2 |
| | *Annotator* | **88.5** | **49.6** | **92.5** | **58.6** | **68.7** | 77.7 | 71.0 | **72.4** |
| ADA | Random | 83.6 | 51.6 | 91.9 | 56.4 | 64.5 | 80.9 | 75.0 | 71.9 |
| | Entropy [21] | 86.2 | **59.2** | 90.2 | 53.9 | 66.3 | 80.3 | 75.5 | 73.1 |
| | Margin [11] | 88.5 | 46.8 | 91.7 | 58.1 | 65.0 | 78.4 | 71.2 | 71.4 |
| | *Annotator* | **88.8** | 58.6 | **93.1** | **62.6** | **68.4** | **82.4** | **77.3** | **75.9** |
| | Target-Only | 89.2 | 73.2 | 95.6 | 71.4 | 75.2 | 87.9 | 85.1 | 82.5 |

Table A5: Per-class results on task of KITTI $\xrightarrow{7}$ nuScene (MinkNet [4]) using only 5 voxel budgets. DA results are reported from [18].

## B.6 Additional qualitative results

In order to provide more qualitative insights, we show the error maps that depict the differences between our model's predictions and the Ground-Truth labels. These error maps are showcased on the KITTI val set and models are trained on the adaptation tasks of SynLiDAR → KITTI (Figure A2) and nuScenes → KITTI (Figure A3), respectively. It is important to emphasize that *Annotator* (ADA) emerges as the top-performer, capitalizing on the advantages of pre-trained models and the presence of annotations in the source domain.

| Model | vehicle | person | road | sidewalk | terrain | manmade | vegetation | mIoU |
|---|---|---|---|---|---|---|---|---|
| Source-Only | 44.1 | 4.3 | 67.6 | 39.4 | 34.9 | 41.2 | 10.5 | 34.6 |
| **DA** Mix3D [14] | 37.9 | 6.7 | 42.0 | 5.7 | 27.6 | 41.2 | 65.4 | 32.4 |
| CoSMix [17] | 44.6 | 13.9 | 36.1 | 10.2 | 29.3 | 54.4 | 69.1 | 36.8 |
| SN [22] | 25.7 | 5.5 | 19.6 | 2.2 | 23.5 | 27.7 | 61.1 | 23.6 |
| RayCast [12] | 28.3 | 16.1 | 45.8 | 9.4 | 20.6 | 38.6 | 61.8 | 31.5 |
| LiDOG [18] | 60.1 | 9.0 | 47.4 | 16.4 | 32.6 | 54.2 | 68.8 | 41.2 |
| **AL** Random | 95.5 | 0.0 | 86.2 | 70.3 | 74.1 | 83.3 | 86.8 | 70.9 |
| Entropy [21] | 96.4 | 0.0 | 84.3 | 68.9 | **75.7** | 82.8 | **87.0** | 70.7 |
| Margin [11] | 94.8 | **35.3** | 83.6 | 68.8 | 65.2 | 80.8 | 83.0 | 73.1 |
| *Annotator* | **96.9** | 33.9 | **86.8** | **73.4** | 73.3 | **86.5** | **87.0** | **76.8** |
| **ASFDA** Random | 94.3 | 0.0 | 85.0 | 67.5 | 72.1 | 83.1 | **86.3** | 69.7 |
| Entropy [21] | 95.6 | 0.0 | 83.7 | 66.5 | 73.1 | 79.8 | 85.0 | 69.1 |
| Margin [11] | 94.1 | **24.6** | 82.1 | 66.4 | 64.2 | 78.8 | 82.1 | 70.3 |
| *Annotator* | **96.6** | 21.9 | **88.5** | **75.7** | **74.1** | **84.2** | 86.1 | **75.3** |
| **ADA** Random | 95.1 | 42.6 | 88.7 | 70.0 | 69.8 | 75.3 | 81.5 | 74.7 |
| Entropy [21] | **96.0** | 32.2 | 86.2 | 69.1 | 70.8 | 79.6 | 84.0 | 74.0 |
| Margin [11] | 94.5 | 59.6 | **89.4** | 69.4 | 70.2 | 74.5 | 79.6 | 76.7 |
| *Annotator* | 95.8 | **66.1** | 88.5 | **74.9** | **75.9** | **84.2** | **86.9** | **81.8** |
| Target-Only | 97.6 | 60.6 | 90.7 | 79.3 | 76.5 | 89.1 | 89.2 | 83.3 |

Table A6: Per-class results on task of nuScenes $\xrightarrow{7}$ KITTI (MinkNet [4]) using only 5 voxel budgets. DA results are reported from [18].

| Model | car | bi.cle | mt.cle | truck | oth-v. | pers. | b.clst | m.clst | road | park. | sidew. | oth-g. | build. | fence | veget. | trunk | terra. | pole | traff. | mIoU |
|---|---|---|---|---|---|---|---|---|---|---|---|---|---|---|---|---|---|---|---|---|
| Source-Only | 67.1 | 6.9 | 22.8 | 0.5 | 5.9 | 30.1 | 56.9 | 4.2 | 18.3 | 6.3 | 31.1 | 0.3 | 30.8 | 11.8 | 63.9 | 29.9 | 42.9 | 25.5 | 4.1 | 24.2 |
| **AL** Random | 92.2 | 0.0 | 10.4 | 25.8 | 18.5 | 23.6 | 0.0 | **0.8** | 85.2 | 23.2 | 69.4 | 1.2 | 83.2 | 44.5 | 86.2 | 56.8 | **73.1** | 54.8 | 27.8 | 40.9 |
| Entropy [21] | **94.4** | **6.1** | **56.2** | **67.9** | **38.6** | **57.2** | **72.4** | 0.0 | 80.3 | 22.2 | 64.1 | **3.2** | 83.6 | 44.5 | **86.4** | 58.7 | 72.7 | 58.7 | 35.0 | 52.7 |
| Margin [11] | 90.4 | 0.0 | 34.2 | 53.9 | 31.2 | 45.8 | 68.1 | 0.0 | 80.5 | 19.4 | 66.8 | 0.2 | 79.3 | 46.3 | 82.3 | 56.8 | 64.4 | 51.5 | 23.2 | 47.1 |
| *Annotator* | 93.9 | 1.6 | 49.9 | 48.9 | 36.4 | 50.2 | 71.8 | 0.1 | **86.2** | 24.6 | 72.3 | 1.4 | **86.6** | **53.1** | 86.4 | 63.1 | 72.5 | **61.7** | **42.8** | **52.8** |
| **ASFDA** Random | 92.8 | 0.0 | 0.0 | 33.3 | 24.6 | 1.1 | 0.0 | 0.0 | 90.0 | 35.7 | 76.1 | 0.0 | 86.9 | 52.5 | 87.1 | 59.3 | 75.4 | 57.9 | 22.9 | 41.7 |
| Entropy [21] | **95.1** | 1.0 | **59.8** | 62.3 | **44.0** | 55.4 | 77.3 | **1.0** | 77.2 | 18.4 | 60.3 | 0.1 | 82.6 | 44.5 | 84.9 | 59.5 | 70.4 | 60.0 | 36.1 | 52.1 |
| Margin [11] | 90.7 | 1.6 | 45.6 | 63.2 | 31.4 | 48.6 | 63.8 | 0.0 | 85.0 | 27.6 | 70.5 | 0.0 | 81.5 | 48.1 | 84.1 | 57.4 | 69.9 | 54.8 | 23.2 | 49.9 |
| *Annotator* | 94.5 | **10.6** | 47.6 | **71.9** | 43.5 | 53.9 | 67.1 | 0.0 | 86.9 | 24.6 | 73.4 | 1.8 | 85.8 | 51.1 | 85.6 | **64.1** | 71.6 | 60.8 | 41.7 | **54.6** |
| **ADA** Random | 92.3 | 10.9 | 40.7 | 42.3 | 28.8 | 50.8 | 71.9 | 0.0 | **88.1** | 27.5 | **73.8** | 2.5 | 84.3 | 49.6 | 83.6 | 59.6 | **69.9** | 54.2 | 38.6 | 51.0 |
| Entropy [21] | 94.2 | **16.1** | 53.3 | **60.1** | 39.2 | **61.4** | 79.8 | 2.2 | 82.4 | 18.6 | 65.4 | 1.4 | 81.7 | 46.1 | 83.8 | 61.0 | 65.2 | 55.1 | 35.3 | 52.8 |
| Margin [11] | 92.1 | 1.0 | **56.9** | 47.5 | 32.1 | 50.7 | **82.8** | 0.0 | 84.5 | 25.5 | 68.5 | 0.2 | 78.3 | 54.0 | 81.7 | 57.8 | 64.8 | 52.2 | 31.8 | 50.7 |
| *Annotator* | **94.7** | 14.8 | 56.7 | 56.8 | **45.3** | 60.4 | 79.0 | 1.3 | 87.3 | **28.6** | 73.0 | 1.8 | **85.4** | 54.3 | 83.9 | 65.2 | 66.5 | 60.0 | 40.9 | **55.6** |
| Target-Only | 96.7 | 25.6 | 73.6 | 81.0 | 61.5 | 73.6 | 90.9 | 0.2 | 93.0 | 46.1 | 79.9 | 0.1 | 89.9 | 58.7 | 86.8 | 67.3 | 71.5 | 65.1 | 48.8 | 63.7 |

Table A7: Per-class results on task of SynLiDAR $\xrightarrow{19}$ KITTI (SPVCNN [20]) with 5 voxel budgets.

| Model | car | bike | pers. | rider | grou. | buil. | fence | plants | trunk | pole | traf. | garb. | cone. | mIoU |
|---|---|---|---|---|---|---|---|---|---|---|---|---|---|---|
| Source-Only | 51.7 | 3.1 | 46.7 | 46.0 | 80.0 | 57.7 | 37.2 | 66.4 | 29.2 | 28.8 | 1.1 | 21.3 | 12.3 | 37.0 |
| **AL** Random | **35.3** | 43.9 | 37.7 | 9.2 | 77.0 | **67.8** | 42.5 | 70.7 | 27.8 | **28.8** | 21.5 | 0.0 | 0.0 | 35.5 |
| Entropy [21] | 24.1 | 35.9 | 35.0 | 22.6 | 78.4 | 61.2 | 42.1 | 71.9 | 14.4 | 22.1 | 15.2 | 16.0 | 18.6 | 35.2 |
| Margin [11] | 33.5 | 41.3 | 55.1 | **47.0** | 78.4 | 54.4 | 36.6 | 62.0 | **41.7** | 27.5 | 23.3 | **20.1** | **31.4** | 42.9 |
| *Annotator* | 31.6 | **44.9** | **56.4** | 46.8 | **78.7** | 65.8 | **50.4** | **73.2** | 32.6 | 26.9 | **36.2** | 16.1 | 23.7 | **44.9** |
| **ASFDA** Random | 38.3 | 48.1 | 44.5 | 16.8 | 76.7 | **68.9** | 46.7 | 71.1 | 20.8 | 30.2 | 29.6 | 0.0 | 0.0 | 37.8 |
| Entropy [21] | 34.5 | 42.7 | 54.4 | 39.4 | 77.6 | 66.6 | 39.7 | 71.3 | 19.0 | 27.5 | 31.5 | 2.3 | 19.9 | 40.5 |
| Margin [11] | 32.7 | 44.1 | 57.6 | 52.2 | **77.9** | 59.3 | 42.8 | 70.3 | **41.6** | **33.4** | 31.4 | **22.9** | 16.4 | 44.8 |
| *Annotator* | **42.8** | **49.3** | **58.7** | **52.9** | 76.2 | 67.4 | **52.7** | **71.4** | 26.9 | 31.5 | **33.7** | 16.0 | **37.6** | **47.5** |
| **ADA** Random | 51.8 | **44.8** | 55.0 | 47.1 | 75.4 | **69.6** | 51.6 | **71.2** | 32.3 | 27.3 | 20.2 | 2.1 | 1.3 | 42.3 |
| Entropy [21] | 54.5 | 42.7 | 65.3 | 57.6 | 79.4 | 60.8 | 55.0 | 70.2 | 29.2 | 28.8 | 18.7 | 5.0 | 40.5 | 46.8 |
| Margin [11] | 36.8 | 30.0 | **65.5** | **58.1** | **81.4** | 65.1 | 44.2 | 70.7 | **37.4** | 31.5 | 25.4 | **35.5** | 30.6 | 47.1 |
| *Annotator* | **64.2** | 40.8 | 62.1 | 55.7 | 77.8 | 67.5 | **57.2** | 70.8 | 31.2 | **35.3** | 28.5 | 24.3 | **46.2** | **50.9** |
| Target-Only | 46.5 | 57.3 | 69.6 | 53.9 | 79.8 | 79.6 | 60.5 | 80.9 | 37.9 | 32.3 | 32.4 | 16.1 | 17.6 | 51.9 |

Table A8: Per-class results on task of SynLiDAR $\xrightarrow{13}$ POSS (SPVCNN [20]) with 5 voxel budgets.

| | Model | vehicle | person | road | sidewalk | terrain | manmade | vegetation | mIoU |
|---|---|---|---|---|---|---|---|---|---|
| | Source-Only | 34.4 | 0.2 | 29.7 | 8.5 | 6.5 | 25.9 | 44.0 | 21.3 |
| AL | Random | 82.8 | 31.9 | 87.6 | 41.2 | 59.3 | **77.8** | **74.8** | 65.0 |
| AL | Entropy [21] | 77.9 | 36.3 | 85.6 | 28.9 | 58.3 | 73.6 | 68.5 | 61.3 |
| AL | Margin [11] | 77.9 | 20.4 | 84.3 | 34.6 | 48.8 | 71.7 | 67.3 | 57.8 |
| AL | *Annotator* | **85.9** | **47.0** | **92.3** | **57.6** | **65.8** | 77.9 | 73.4 | **71.4** |
| ASFDA | Random | 83.8 | 34.9 | 88.9 | 45.4 | 59.5 | 79.3 | **76.5** | 66.9 |
| ASFDA | Entropy [21] | 87.6 | 34.4 | 87.4 | 36.6 | 60.1 | **80.3** | 75.7 | 66.0 |
| ASFDA | Margin [11] | 77.6 | 25.1 | 85.4 | 40.6 | 54.8 | 70.9 | 67.7 | 60.3 |
| ASFDA | *Annotator* | **88.4** | **46.8** | **92.8** | **58.7** | **66.4** | 78.1 | 73.8 | **72.1** |
| ADA | Random | 84.1 | 27.1 | 91.0 | 53.0 | 55.1 | 72.1 | 67.9 | 64.3 |
| ADA | Entropy [21] | **89.7** | 44.2 | 89.8 | 51.3 | 53.7 | 71.7 | 63.7 | 66.3 |
| ADA | Margin [11] | 85.4 | 17.1 | 91.8 | 55.0 | 55.3 | 72.1 | 65.8 | 63.2 |
| ADA | *Annotator* | 89.5 | **50.2** | **92.1** | **57.5** | **66.3** | **78.6** | **72.2** | **72.3** |
| | Target-Only | 93.1 | 71.7 | 94.6 | 66.3 | 72.1 | 87.0 | 84.0 | 81.3 |

Table A9: Per-class results on task of KITTI $\xrightarrow{7}$ nuScene (SPVCNN [20]) with 5 voxel budgets.

| | Model | vehicle | person | road | sidewalk | terrain | manmade | vegetation | mIoU |
|---|---|---|---|---|---|---|---|---|---|
| | Source-Only | 65.7 | 44.5 | 56.2 | 32.6 | 30.5 | 53.2 | 47.0 | 47.1 |
| AL | Random | 94.2 | 0.0 | 84.7 | 68.1 | 73.9 | 84.4 | 87.6 | 70.4 |
| AL | Entropy [21] | 94.2 | 13.0 | 79.0 | 60.4 | 73.1 | 80.8 | 85.8 | 69.5 |
| AL | Margin [11] | 93.6 | 29.4 | 84.0 | 69.9 | 64.4 | 81.4 | 83.5 | 72.3 |
| AL | *Annotator* | **96.7** | **48.2** | **87.9** | **74.6** | **75.8** | **85.8** | **87.8** | **79.5** |
| ASFDA | Random | 94.3 | 3.5 | 81.6 | 60.8 | 68.8 | 81.9 | 85.7 | 68.1 |
| ASFDA | Entropy [21] | 95.1 | 11.0 | 77.2 | 55.6 | 69.5 | 78.8 | 84.5 | 67.4 |
| ASFDA | Margin [11] | 92.5 | 32.9 | 85.0 | 70.5 | 65.2 | 81.6 | 83.1 | 73.0 |
| ASFDA | *Annotator* | **96.7** | **55.4** | **87.7** | **75.3** | **75.6** | **85.5** | **87.5** | **80.5** |
| ADA | Random | 94.1 | 39.5 | **88.9** | **73.0** | 71.3 | 80.0 | 83.4 | 75.8 |
| ADA | Entropy [21] | 90.2 | **63.6** | 84.2 | 70.9 | 65.9 | 66.5 | 66.6 | 72.6 |
| ADA | Margin [11] | 92.7 | 58.6 | 88.2 | 71.0 | 69.3 | 71.4 | 75.3 | 75.3 |
| ADA | *Annotator* | **95.0** | 55.0 | 86.5 | 69.0 | **74.9** | **82.8** | **86.0** | **78.4** |
| | Target-Only | 98.0 | 71.7 | 91.0 | 79.9 | 74.6 | 90.5 | 89.0 | 85.0 |

Table A10: Per-class results on task of nuScenes $\xrightarrow{7}$ KITTI (SPVCNN [20]) with 5 voxel budgets.

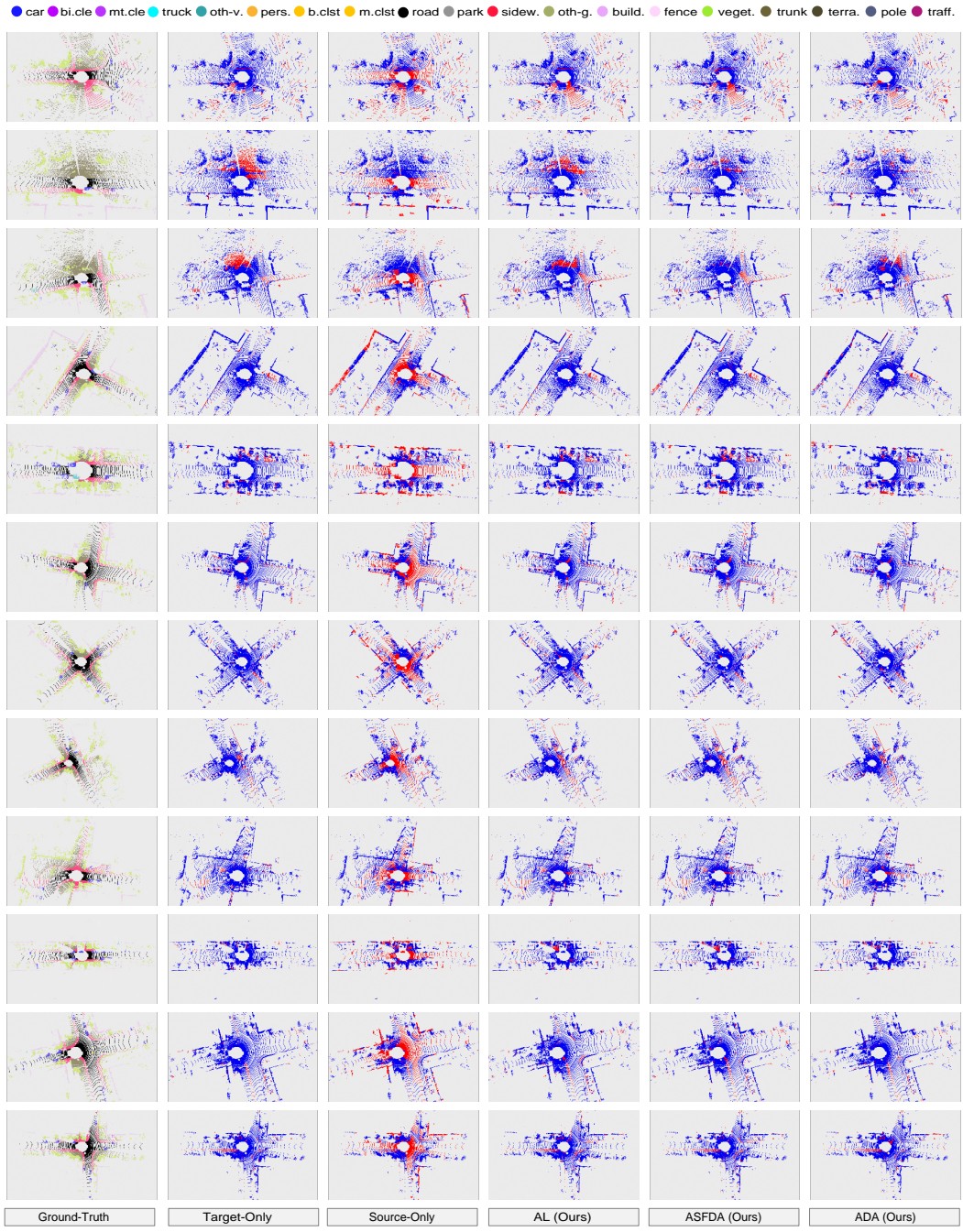

Figure A2: Visualization of error maps for the task SynLiDAR $\xrightarrow{19}$ KITTI (MinkNet [4]). From left to right: Ground-Truth, Target-Only, Source-Only, our Annotator under AL, ASFDA, and ADA are shown one by one. The correct and incorrect predictions are painted in blue and red to highlight the differences. Best viewed in color.

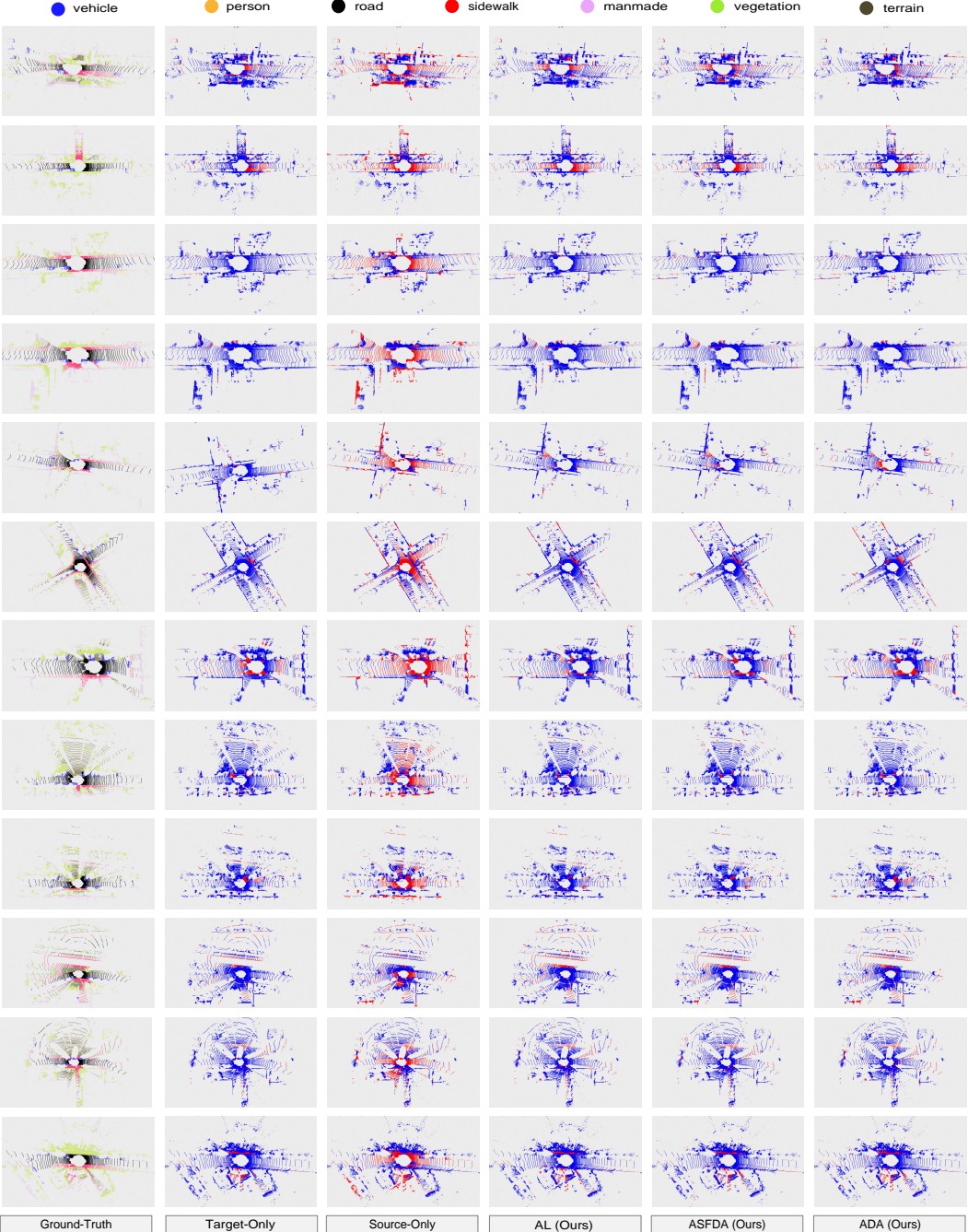

Figure A3: Visualization of error maps for the task nuScenes $\xrightarrow{7}$ KITTI (MinkNet [4]). From left to right: Ground-Truth, Target-Only, Source-Only, our Annotator under AL, ASFDA, and ADA are shown one by one. The correct and incorrect predictions are painted in blue and red to highlight the differences. Best viewed in color.

## C   Public Resources Used

We acknowledge the use of the following public resources, during the course of this work:

- SynLiDAR[1] .......................................................... MIT License
- SemanticKITTI[2] ............................................... CC BY-NC-SA 4.0
- SemanticKITTI-API[3] ................................................. MIT License
- SemanticPOSS[4] ............................................... CC BY-NC-SA 3.0
- nuScenes[5] .................................................... CC BY-NC-SA 4.0
- nuScenes-devkit[6] ........................................... Apache License 2.0
- Minkowski Engine[7] ................................................. MIT License
- SPVNAS[8] ........................................................ MIT License
- PCSeg[9] ...................................................... Apache License 2.0
- LaserMix[10] .................................................. CC BY-NC-SA 4.0
- GIPSO[11] ...................................... GNU General Public License v3.0
- SalsaNet[12] ....................................................... MIT License
- PolarNet[13] ................................................ BSD 3-Clause License
- RIPU[14] .......................................................... MIT License