# OpenReview forum: "Annotator: A Generic Active Learning Baseline for LiDAR Semantic Segmentation"
_NeurIPS.cc/2023/Conference — NeurIPS 2023 poster_

### Official Review · Reviewer_KrDz · 2023-06-29

**Soundness:** 3 good
**Presentation:** 2 fair
**Contribution:** 1 poor
**Rating:** 6
**Confidence:** 3

**Summary:**

This work benchmarks several point selection approaches for label-efficient LiDAR point cloud semantic segmentation. Their proposed criterion leads to better results than existing selection approaches and good generalization with few annotations. They use several settings depending on the accessibility of auxiliary data to overcome the "cold-sart" problem.

**Strengths:**

- the proposed criterion is sensible and leads to good results

- the experiments are extensive

- the definition of three different setting (no auxiliary data, pretrained model, auxiliary annotated data) is helpful

- active learning is a critical problem for the industrialization of ML methods and is not studied nearly enough

**Weaknesses:**

- the entire contribution boils down to the "voxel confusion degree" selection criterion. Yet the authors present the entire field of active learning as a contribution

- the quality of writing is quite low, with many misused technical terms (softmax cross-entropy) and vague statements (active learning [is] an optimum paradigm).

- the authors only compared their approach to 2 selection criteria [27,76], but could have used many other approaches, such as  [25, 35, 58, 59, 78, 84]. Otherwise, they need to explain why these approaches do not apply

- The results seem dubious. Annotating only 5 voxels (how many points total?) leads to such strong performance in a setting with many classes. Even when using 40 million parameters- networks trained from scratch and no particular measures taken to avoid the extreme overfitting bound to occur?

**Questions:**

Q1) How do you explain that large networks can be trained with cross entropy and only a handful of points? How many epochs are you using?

Q2) Why not compare to any of the other DA approaches  [25, 35, 58, 59, 78, 84]

S1) proof read the entire text looking for mistakes, vague and imprecise statements.

**Limitations:**

not given

---

> ### Author Rebuttal · Authors · 2023-08-09
>
> We appreciate the comments from the reviewer KrDz. We have answered all the questions and sincerely hope they can address the concerns.
>
> **Q1: the entire contribution boils down to the "voxel confusion degree" selection criterion. Yet the authors present the entire field of active learning as a contribution.**
>
> *Motivation*: Recently, there has been significant interest in achieving label-efficient LiDAR semantic segmentation. Both AL within the same data distribution and DA across different data distributions hold promise as solutions to alleviate the manual annotation burden. Nonetheless, **the absence of a standardized baseline** could hinder the advancement of research. Further, AL coupled with DA has great practical significance, which has been explored in 2D image classification. Yet, **it is still unclear how to make it work well in 3D domains**.
>
> *Purpose*: This paper aims to present a simple and general active learning baseline for LiDAR semantic segmentation via the voxel-centric selection.  With its simplicity and strong performance across various backbones (e.g., MinkNet, SPVCNN, SalasNet, PolarNet, etc.), regardless of in distribution or out of distribution setting (i.e., AL, ASFDA, and ADA), and robustness over simulation-to-real and real-to-real scenarios, we hope this baseline can facilitate future research.
>
> *Contribution*: We would like to emphasize that the main contributions lie in three aspects.
> - a label acquisition strategy (VCD) is **more robust and diverse** to select samples efficiently under a domain shift;
> - a voxel-centric online active learning can largely reduce the labelling cost of enormous point clouds. Particularly, only requiring **1,000x fewer annotations** can reach a satisfactory performance;
> - **generally applicable** for various network architectures (voxel-, range-, and BEV-view, etc), settings (in distribution and out of distribution), and scenarios (simulation-to-real and real-to-real).
>
> **Q2: the quality of writing is quite low, with many misused technical terms (softmax cross-entropy) and vague statements (active learning [is] an optimum paradigm)**
>
> Thank you. We will fix minor mistakes and revise paper thoroughly.
>
> **Q3: the authors only compared their approach to 2 selection criteria [27,76], but could have used many other approaches, such as [25, 35, 58, 59, 78, 84]. Otherwise, they need to explain why these approaches do not apply.**
>
> This question is very interesting and valuable. **First**, we would like to clarify that our goal is to build a simple and general baseline, in which we benchmark several selection strategies, i.e., Random, Entropy[76], and Maigin[27] since they are typically used to testify the proposed selection strategy. **Second**, as mentioned before, there is no standardized baseline. Thus, it might be unfair to compare with some works across different scenarios and settings. Here, we compare our method with other works [25, 35, 58, 59, 78, 84] as much as possible on the basis of the benchmarks presented in this paper. These works can be categorized into three groups:
>
> 1. *active learning for 2D image classification[58, 84].* core-set[58] proposes a diversity-based sample selection strategy for image classification, where a clustering algorithm is adopted and DUC[84] estimates uncertainty from the perspective of Dirichlet distribution. We modify these two selection strategies into our baseline and compare with them on two simulation-to-real tasks based on MinkNet. The detailed results are reported in `Table R2 of the one-page PDF`. Please be aware that our method continues to deliver the most favorable outcomes in two case.
>
> 2. *active learning for 3D LiDAR semantic segmentation[25, 35, 59].* Lidal [25] proposed a frame-level selection strategy. Less [35] and SSDR-AL [59] first introduced a pre-segment stage (a heuristic algorithm in [35] and superpoint in [59]) and then conducted labeling process on outdoor and indoor scenes respectively.
>
> We first compare Lidal and Less on outdoor KITTI dataset as follows.
>
> mehotd|budget|MinkNet|SPVCNN
> -|-|-|-
> Lidal|1%|47.5|48.5
> Less|0.1%|52.8|51.1
> Ours|0.1%|53.7|52.8
>
> Our method is simpler yet achieves better performance. Then, following [59], we conduct experiment on indoor S3DIS dataset. Detailed results and analyses can be found at response of Q6 for Reviewer dFZX.
>
> 3. *Squeezeseg [78]* is an end-to-end road-object segmentation method with CNNs and CRFs and synthetic data (GTA-V) is used to train the model. It is not applicable to our setting.
>
> **Finally**, we would like to emphasize that your suggestion is very helpful, and we will pay attention to it when revising the paper.
>
> **Q4: the results seem dubious. Annotating only 5 voxels (how many points total?) leads to such strong performance in a setting with many classes. Even when using 40 million parameters-networks trained from scratch and no particular measures taken to avoid the extreme overfitting bound to occur?**
>
> Thanks for the valuable question. On average, the total number of annotated points (5 voxels) in a frame (64,000 points in total) is about 35. `Figure R2 of the one-page PDF` provides curves of training loss and validation loss on task of SynLiDAR->POSS under AL (train from scratch) and ASFDA (train from an auxiliary model) settings. Both models are based on MinkNet (21.7 MB). The training loss and validation loss have been steadily decreasing and the final validation loss is smaller than the training loss as well, verifying that there is no extreme overfitting. Another finding is that the validation loss of ASFDA is smaller than that of AL, which confirms the power of auxiliary model. We will add this analysis in the revision.
>
> **Q5: How many epochs are you using?**
>
> The pre-training process uses 10 epochs and others (AL/ASFDA/ADA) use 50 epochs.
>
> **Q6: limitations: not given**
>
> Actually, we have discussed the limitations in the Conclusion. We will make it more clear in a separate section.

---

> > ### Comment · Reviewer_KrDz · 2023-08-11
> > **Thank you**
> >
> > The authors have addressed all my queries and reservations.
> >
> > While I initially perceived the contribution as too limited, several points have swayed my opinion:
> > - the authors make a good point relative to the lack of standardized baseline, and their method makes a good stepping stone in that direction.
> > - they added more comparisons that still show the merits of their approach.
> > - the method, while simple, yields superior results. This makes its simplicity an asset.
> >
> > I also recognize that some of my earlier concerns arose from my own minsunderstanding (specifically regarding the 5 voxels equating to 35 points).
> >
> > Considering the above factors, I am now leaning towards endorsing this paper and have subsequently revised my rating.

---

> > > ### Author Response · Authors · 2023-08-13
> > > **Thanks for your suggestions**
> > >
> > > We really appreciate your comments and reply. We believe that our work can establish a simple and general baseline for label-efficient LiDAR semantic segmentation, and hope our observation can inspire more subsequent label-efficient works.

---

### Official Review · Reviewer_a35W · 2023-07-03

**Soundness:** 3 good
**Presentation:** 3 good
**Contribution:** 3 good
**Rating:** 6
**Confidence:** 4

**Summary:**

This paper introduced a baseline for active learning called Annotator for LiDAR point cloud semantic segmentation. The paper includes an analysis of various active selection strategies, including random selection, entropy-based selection, margin-based selection, and a novel strategy called voxel confusion degree (VCD). The study investigates three different setups: active learning (AL), active source-free domain adaptation (ASFDA), and active domain adaptation (ADA). Through experiments conducted on multiple Simulation-to-Real and Real-to-Real benchmarks, the proposed method demonstrates its effectiveness.

**Strengths:**

1.	The paper is well-written and easy to follow.
2.	Considering the significant challenge of annotating point-wise point cloud data in LiDAR segmentation, exploring active learning approaches for this task is valuable and important.
3.	The idea of annotator is simple but effective.
4.	The authors conducted comprehensive benchmarks across different setups, including AL, ASFDA, and ADA. This extensive evaluation covers several novel investigations that have not been explored before.

**Weaknesses:**

1. This paper lacks in-depth insights and analysis. What is the fundamental intuition behind it? Why do these active selection strategies contribute to performance gains? Furthermore, what sets the proposed VCD apart and enables it to achieve superior performance improvements?
2. The selection of voxel grids plays a crucial role in determining the final performance, particularly when it comes to random selection. However, this paper did not include error bars in relation to the random seed for multiple experiments.

**Questions:**

1. Why did the annotator achieve significantly smaller performance gains in SynLiDAR->KITTI in Table 2?
2. A minor issue: The per-class IoU performance of PolarMix is provided in its supplementary material, which is missing in Tables 2 and 3.
3. Please also refer to the weakness.

**Limitations:**

This paper discusses the limitations of their work in the conclusion. However, it is suggested to have a separate section dedicated specifically to addressing these limitations.

---

> ### Author Rebuttal · Authors · 2023-08-09
>
> We sincerely thank the reviewer a35W for the very constructive comments. We are glad that the reviewer acknowledge that the task is valuable and important, the idea is simple but effective, and the investigation is novel. Here we address the biggest concern raised by the reviewer, i.e., the fundamental intuition behind voxel-centric selection, and hope our response can address this concern.
>
> **Q1: This paper lacks in-depth insights and analysis. What is the fundamental intuition behind it?**
>
> Thanks for the valuable question. We would clarify that the intuition of this paper is to establish a simple and general baseline for label-efficient LiDAR semantic segmentation. To mitigate the annotation burden in model training, most of existing paradigms delves into active learning (in distribution) or domain adaptation (out of distribution), separately. However, there are **not unified setting and baseline**, which hinders the research. Indeed, the two classes of approaches are deeply intertwined, but **little work has been done to consider the combination in 3D domains**. This paper targets at delivering a simple and general online active learning baseline by a voxel-centric selection. It unifies active learning and domain adaptation for LiDAR semantic segmentation. Moreover, sufficient and convincing empirical results provide insights for label-efficient 3D applications.
>
> **Q2: Why do these active selection strategies contribute to performance gains?**
>
> At first, we would like to clarify that there exist image/frame-, region-, and point-based selection strategies in previous research. Active learning aims to identify informative instances for labeling via a selection strategy. Thus, given limited budget, the selection strategy can select the most informative instances for labeling, which can be used to train the model, leading to performance gains.
>
>
> **Q3: What sets the proposed VCD apart and enables it to achieve superior performance improvements?**
> Our VCD design is inspired by an observation: traditional AL setting often focuses on learning a model from scratch rather than adapting under domain shift. In practice, models are trained in an auxiliary domain and deployed in a new domain of interest. In this case, existing AL strategies are less effective since uncertainty estimates on the new domain may be miscalibrated. By contrast, our label acquisition strategy (VCD) is tailored to estimate category diversity instead of uncertainty within a voxel, which is more robust under domain shift. Specifically, we first generate pseudo label for each point within a voxel based on model prediction. Second, we count the percentage of each category within the voxel. Third, we take the entropy of the statistical information within the voxel as a score. Finally, we select voxels with high score. The **higher the score, the more predicted classes within a voxel**, and we think this would help to train the model after being annotated. As shown in Fig. 5, we clearly see that the true class distribution of SemanticPOSS is exactly a long-tail distribution while our method can pick out more voxels that contain tail classes. Also, per-class results in Table 3-4 of the main paper and `Table R1 of the one-page PDF` show the superiority of our method in both tail and majority classes.
>
>
> **Q4: The selection of voxel grids plays a crucial role in determining the final performance, particularly when it comes to random selection. However, this paper did not include error bars in relation to the random seed for multiple experiments.**
>
> It is a very interesting question regarding the viability of Annotator in the real world. Actually, current results in our experimental section are the mean accuracy across three runs with different random seeds, and we include error bars on task of SynLiDAR $\to$ KITTI using three random runs in `Figure R1 of the one-page PDF`. And we will add the error bars on all benchmarks in the revision.
>
>
> **Q5: Why did the annotator achieve significantly smaller performance gains in SynLiDAR->KITTI in Table 2?**
>
> This question is very interesting and valuable. Please note that the performance gain (0.1% under AL/ 2.5% under ASFDA/ 2.8% under ADA in SynLiDAR $\to$ KITTI in Table 2) is compared to the state-of-the-art baseline method. We think that the numerical improvement should attribute to several reasons. First, there includes challenging variations in data distribution and annotation quality between synthetic SynLiDAR dataset and real-world KITTI dataset. Second, for simplicity, we keep all hyperparameters unchanged when change the backbone from MinkNet to SPVCNN.
>
>
> **Q6: A minor issue: the per-class IoU performance of PolarMix is provided in its supplementary material, which is missing in Tables 2 and 3.**
>
> We apologize for missing per-class IoU performance of PolarMix. Here we report them in the following tables and will add them in the revision.
>
> SynLiDAR $\to$ KITTI (MinkNet)
>
> mehotd|car|bi.cle|mt.cle|truck|oth-v.|pers.|b.clst|m.clst|road|park.|sidew.|oth-g.|build.|fence| veget.|trunk|terra.|pole|traf.|mIoU
> -|-|-|-|-|-|-|-|-|-|-|-|-|-|-|-|-|-|-|-|-
> PolarMix|76.3|8.4|17.8|3.9|6.0|26.6|40.8|15.9|70.3|0.0|44.4|0.0|68.4|14.7|69.6|38.1|37.1|40.6|10.6|31.0
>
> SynLiDAR $\to$ POSS (MinkNet)
> mehotd|pers.|rider|car|trunk|plants|traf.|pole|garb.|buil.|cone|fence|bike|grou.|mIoU
> -|-|-|-|-|-|-|-|-|-|-|-|-|-|-
> PolarMix|32.6|39.1|25.0|11.9|64.2|5.8|29.6|15.3|44.8|13.3|23.8|10.7|79.0|30.4
>
> Compared with PolarMix, a strong DA method, we observe that traditional AL is more powerful than DA as it allows querying limited data to be labeled. This paper shows that, given limited annotations, DA (i.e., ASFDA & ADA) can achieve on-par performance with fully-supervised counterparts.
>
> **Q7: Limitations: it is suggested to have a separate section dedicated specifically to addressing these limitations.**
>
> Thanks for the valuable suggestion. We will add a separate section to discuss the limitations in the revision.

---

> > ### Comment · Reviewer_a35W · 2023-08-18
> > **Thanks for the response**
> >
> > The response has addressed most of my concerns. Given that the authors emphasize the significance of this work as a baseline for future studies, could you please provide information regarding the potential release of the code? If so, when is the expected release date?

---

> > > ### Author Response · Authors · 2023-08-18
> > > **Thank you for your positive feedback on our response.**
> > >
> > > Dear Reviewer a35W,
> > >
> > > We're glad to hear that your concerns have been addressed. We have included the code for review in the preliminary submission. In terms of releasing the code, we are actively working to prepare the codebase for public distribution. We understand the value of reproducibility in research and are committed to making the code available to the community.
> > >
> > > At this stage, we expect to release the code within the next month. We will ensure that the code is well documented and easy to use, so that other researchers can build on our work and continue to contribute to the field. We appreciate your interest in our code release and will keep you updated on its progress.
> > >
> > > Thank you for your understanding and continued interest in our research.
> > >
> > > Best regards,
> > >
> > > Submission2070 Authors.

---

> > > > ### Comment · Reviewer_a35W · 2023-08-21
> > > > **Thanks for the information**
> > > >
> > > > I appreciate your future plan for releasing the code. I will raise my rating to weak accept.

---

### Official Review · Reviewer_dFZX · 2023-07-05

**Soundness:** 2 fair
**Presentation:** 3 good
**Contribution:** 2 fair
**Rating:** 6
**Confidence:** 4

**Summary:**

This work proposes a general and efficient data annotation pipeline, namely Annotator, to label LiDAR data for semantic segmentation. Specifically, the proposed method introduces a voxel-centric online selection strategy to determine which voxel should be annotated by humans. Voxel confusion degree (VCD) is thus proposed based on the classification labels predicted by a semantic segmentation network. To prove its efficiency, the authors also conduct some experiments on domain adaption and they use 1,000 times fewer annotations.

**Strengths:**

This work proposes a new criterion namely voxel confusion degree (VCD) to evaluate the certainty of a voxel. It seems using this criterion the authors only need very few times of annotations to achieve better performance.

The experiments support the claim of this paper: the proposed method uses the least annotations and achieves better performance.

**Weaknesses:**

1. In Fig. 2 (ii), it is ASFDA source-free domain adaptation but in (1) pre-trained with source data. This is a bit confusing.

2. The VCD is well defined. How to use it to determine the next annotated voxel is not clear. Should we choose a voxel with larger VCD? Why is that? The motivation or insights are missing.

3. Algorithm 1 is not very informative but more like active learning pipeline.

4. The proposed method Annotator can be only applied to voxelized data. How does the size of voxels affect the final annotation rounds?

5. The compared baselines are all voxel based annotation. How about labeling points in BEV? In other words, there would be many ways to annotate LiDAR data. Why voxel is the best? Or is the proposed method best suitable for voxel based partition?

**Questions:**

1 whether the proposed method can be applied to indoor semantic segmentation?

2. Why choose voxel based partition?

3. In Tables 1, 2, 3 The oracle performance (full annotations) should be provided for comparisons.

4. Can the proposed method be applied to other tasks, such as detection or instance segmentation?

**Limitations:**

The authors have addressed the concerns properly.

---

> ### Author Rebuttal · Authors · 2023-08-09
>
> We sincerely thank the reviewer dFZX for the very valuable comments. We are glad that the reviewer acknowledges our new criterion can achieve better performance with only very few times of annotations. Here we address the concerns of our paper, and hope our response can address the concern.
>
> **Q1: In Fig. 2 (ii), it is ASFDA source-free domain adaptation but in (1) pre-trained with source data. This is a bit confusing.**
>
> We apologize for the misunderstanding. One thing that can be confirmed is that only the source model is available in source-free domain adaptation. Actually, Fig. 2 illustrates both (1) source model pre-training process and (2) adaptation and active learning process (ii). Therefore, in (1) we assume that source data is only used to pre-train the source model, while it is not available for adapting the model in (2).
>
> **Q2: The VCD is well defined. How to use it to determine the next annotated voxel is not clear. Should we choose a voxel with larger VCD? Why is that? The motivation or insights are missing.**
>
> Yes, a voxel with larger VCD will be chosen. Recall that, we first obtain pseudo label $\hat{y_i}$ of each point $x_i$ and then divide points within a voxel $v_j$ into $K$ clusters: $v_j^{<k>}=\{x_i\in v_j,\hat{y_i}=k\}$. At the moment, we can collect statistical information about the categories in a voxel. Next, we take the entropy calculated on the percentage of each distinct class as the voxel confusion degree of each voxel, that is, VCD($v_j$)$=-\sum_{k=1}^{K}\frac{|v_j^{<k>}|}{v_j}\log \frac{|v_j^{<k>}|}{v_j}$, where $|\cdot|$ denotes the number of points in a set. The insight is that if there are many predicted classes within a voxel, we assume that it can help to train the model after being annotated. As shown in Fig. 5, we clearly see that the true class distribution of SemanticPOSS is exactly a long-tail distribution while our method can pick out more voxels that contain tail classes. Also, per-class results in Table 3-4 of the main paper and `Table R1 of the one-page PDF` show the superiority of our method in both tail and majority classes.
>
> **Q3: Algorithm 1 is not very informative but more like active learning pipeline.**
>
> Good suggestion. Most of the lines in Algorithm 1 describe active learning pipeline, to which we add additional settings, namely ASFDA and ADA (line 2), and show how to perform voxel-centric selection and model training (lines 5-6). Perhaps we can describe our algorithm in text.
>
> **Q4: The proposed method Annotator can be only applied to voxelized data. How does the size of voxels affect the final annotation rounds?**
>
> This question is very interesting and valuable. The voxel size $\Delta$ is default set to 0.25 for selection process and 0.05 for training process. In the case of SemantcPOSS, for example, there are about 30,000 voxels per frame. Five rounds of selection are conducted, one voxel per round, resulting in a 1,000x reduction in annotations. When the voxel size is small, the number of voxels will be large, which will increase selection rounds. Here, we conduct experiments on different size (from 0.05 to 0.35) of voxels for selection process. The following results are conducted on SynLiDAR $\to$ POSS based on MinkNet.
>
> $\Delta$|0.05|0.1|0.15|0.2|0.25|0.3|0.35
> -|-|-|-|-|-|-|-
> \# voxel per frame|64973|54543|43795|36091|30414|25992|22539|
> \# selection rounds|11|9|7|6|5|4|4|
> mIoU in AL|39.6|42.9|43.9|44.2|44.9|45.1|44.8
> mIoU in ASFDA|40.0|46.2|46.0|48.0|48.2|48.3|48.0
>
> It can be seen that the performance of the large voxel grid ($\Delta$>0.2) is more robust to the noise and sparsity of the point clouds. We will add the results to the revision.
>
> **Q5: The compared baselines are all voxel based annotation. How about labeling points in BEV? In other words, there would be many ways to annotate LiDAR data. Why voxel is the best? Or is the proposed method best suitable for voxel based partition?**
>
> Thanks for the valuable question. At first, we would like to clarify that there exist image/frame-, region-, and point-based selection strategies in previous research. The first two require an offline stage, which may be infeasible at large scales. The last one is costly due to the sparsity of point clouds. By contrast, voxel-based selection aims to query the salient and exemplar regions and annotate all points within the region, which is more efficient and flexible. It can be easily applied to different methods, e.g., point-, voxel-, range-, BEV-view. Except for the results of voxel-view method reported in the main paper, we have added experimental results on a range-view method SalsaNext and a BEV-view method PolarNet in `Table R1 of the one-page PDF`. It is seen that our method still brings a large improvement using either range- or BEV-view method with a limited budget.
>
> **Q6: whether the proposed method can be applied to indoor semantic segmentation?**
>
> Yes, our method can be applied to indoor semantic segmentation. Following SSDR-AL [59], we have conducted experiments on S3DIS dataset. We report results of comparing the percentage of labeled points required to achieve 90% accuracy for different methods based on Randlanet as follows.
>
> method|Random|Entropy|Margin|SSDR-AL [59]|Ours
> -|-|-|-|-|-
> budget|40.9%|46.7%|43.0%|11.7%|9.9%
>
> It is observed that our method can annotate 1.8% fewer points than SSDR-AL in achieving the 90% performance of fully-supervised counterpart.
>
> **Q7: In Tables 1, 2, 3 The oracle performance (full annotations) should be provided for comparisons.**
>
> Actually, "Target-only" in Tables 1, 2, 3, 4 denotes the oracle performance (full annotations in the target domain). Compared with the oracle performance, we find that our Annotator with 5 voxels per frame being labeled can obtain on-par or better performance.
>
> **Q9: Can the proposed method be applied to other tasks, such as detection or instance segmentation?**
>
> This question is very interesting and valuable. Please ref to `response to Q3 for reviewer zYAC`.

---

### Official Review · Reviewer_N28Y · 2023-07-05

**Soundness:** 2 fair
**Presentation:** 2 fair
**Contribution:** 2 fair
**Rating:** 6
**Confidence:** 4

**Summary:**

This paper presents Annotator, a general and efficient active learning baseline for LiDAR semantic segmentation, which can adapt to different settings and scenarios with minimal annotation cost. Annotator consists of a voxel-centric online selection strategy that exploits the local topology and structure of point clouds to query the most informative voxel grids for annotation. Annotator can also leverage an auxiliary model to address the cold start problem. The paper evaluates Annotator on two datasets (SynLiDAR and SemanticKITTI) with two network architectures (MinkNet and SPVCNN), and shows that Annotator can achieve on-par performance with the fully supervised counterpart using 1000 fewer annotations and outperform existing methods under various active learning settings. The paper states that Annotator is a simple and general solution for label-efficient 3D perception.

**Strengths:**

**Originality**: The paper proposes 3 ideas,  a voxel-centric online active learning baseline, a label acquisition strategy (VCD), and is generally applicable and works for different network architectures.

**Quality**: The paper provides a thorough **experimental evaluation** of Annotator on several simulation-to-real and real-to-real LiDAR semantic segmentation tasks, using different network architectures and baselines. The paper also conducts **ablation studies** to analyze the impact of different components of Annotator, such as voxel size, selection strategy, and auxiliary model. The paper demonstrates that Annotator can achieve **on-par or superior performance** with the fully supervised counterpart using 1000 fewer annotations, and significantly outperform other state-of-the-art methods.

**Clarity**: The paper also provides sufficient background information and related work to situate the contribution of Annotator in the context of existing literature on LiDAR perception, active learning, and domain adaptation. The Supplementary Material is well-written and contains enough details and information as needed.

**Significance**: The paper addresses an important and challenging problem of label-efficient LiDAR semantic segmentation, which has many applications in autonomous driving, robotics, and 3D scene understanding.

**Weaknesses:**

**Major Issues:**
- **Insufficient novelty and contribution**: this paper perform an analysis of several common selection strategies, e.g., Random, Entropy and Margin, which are all proposed in previous research. The newly proposed VCD strategy lacks justification for its design, and contains too few details (L195-L199). The pipelines of distinct active learning settings (e.g., AL, ASFDA, and ADA) seem natural and basic.
- **Insufficient results for experiments:** Although the authors state in the main text, "***Experimentally***, we find that the large voxel grid is also more robust to the noise and sparsity of the point clouds", they provide no experimental results of different $\Delta$ in their main text and Supplementary Material.
- **Insufficient justifications**: For example, about  **Voxel-centric** selection, some justifications are missing in this paper. Why choosing voxel-based selection rather than other methods, e.g., point-based, range-image-based, BEV, etc (L101-102)? Any advantages and limitations of **Voxel-based** methods?

**Minor Issues:**
- **Readability**: Some figures and text in figures (e.g., Figure 6 and Figure A3-A4) are too small for readability. Authors should attach more detailed images/visualization to the supplementary material.
- **Variables undefined** although they are obvious in meaning: e.g., $X_t$ in L155, $X_s$ and $Y_s$ in L157.

**Questions:**

- Is the term '*cell*' (L190) the same meaning as '*voxel*' (L189)? If they're interchangeable, it might be better to use the same word '*voxel*'; if not, it's a good idea to explain their differences, relations, or the definition of '*cell*'.


**Limitations:**

Authors should be rewarded that the limitations and potential negative societal impact are explicitly mentioned in the Conclusion.

---

> ### Author Rebuttal · Authors · 2023-08-09
>
> We sincerely thank the reviewer N28Y for the appreciative and constructive comments. We are encouraged that the reviewer found our three original ideas, thorough experiments, sufficient background, and well-written supplementary material. Here, we would like to respond to the issues and hope that our responses will address the concern.
>
> **Q1: Insufficient novelty and contribution**
>
> We would like to emphasize that the main contributions lie in three aspects.
> - a label acquisition strategy (VCD) is **more robust and diverse** to select samples efficiently under a domain shift;
> - a voxel-centric online active learning can largely reduce the labelling cost of enormous point clouds. Particularly, only requiring **1,000x fewer annotations** can reach a close performance to the fully-supervised counterpart;
> - **generally applicable** for different network architectures (voxel-, range-, and BEV-view, etc), settings (in distribution and out of distribution), and scenarios (simulation-to-real and real-to-real).
>
> **Q2: this paper perform an analysis of several common selection strategies, e.g., Random, Entropy and Margin, which are all proposed in previous research.**
>
> Indeed, selection strategies such as Random, Entropy, and Margin typically serve as baseline methods to verify the effectiveness of the proposed method. In this work, we introduce a novel label acquisition strategy (VCD) and a voxel-centric online active learning. Therefore, we first testify common strategies in our voxel-centric selection (the response of Q6 gives justifications). And then we compare VCD with these strategies to show its superiority. In other words, these common selection strategies
> are not technical contributions of the paper but are necessary to form our baseline.
>
>
> **Q3: The newly proposed VCD strategy lacks justification for its design, and contains too few details (L195-L199).**
>
> Our VCD design is inspired by an observation: traditional AL setting often focuses on learning a model from scratch rather than adapting under domain shift. In practice, models are trained in an auxiliary domain and deployed in a new domain of interest. In this case, existing AL strategies are less effective since uncertainty estimates on the new domain may be miscalibrated. By contrast, our label acquisition strategy (VCD) is tailored to estimate category diversity instead of uncertainty within a voxel, which is more robust under domain shift. Specifically, we first generate pseudo label for each point within a voxel based on model prediction. Second, we count the percentage of each category within the voxel. Third, we take the entropy of the statistical information within the voxel as a score. Finally, we select voxels with high score. The **higher the score, the more predicted classes within a voxel**, and we think this would help to train the model after being annotated.
>
> **Q4: The pipelines of distinct active learning settings (e.g., AL, ASFDA, and ADA) seem natural and basic.**
>
> This question is very interesting and valuable. To mitigate the annotation burden in model training, most of existing paradigms delves into active learning (in distribution) or domain adaptation (out of distribution), separately. However, there are **not unified setting and baseline**, which hinders the research. Indeed, the two classes of approaches are deeply intertwined, but **little work has been done to consider the combination in 3D domains**. This paper targets at delivering a simple and general online active learning baseline by a voxel-centric selection. It unifies active learning and domain adaptation for LiDAR semantic segmentation. Moreover, sufficient and convincing empirical results validate the necessity of these distinct settings.
>
> **Q5: Insufficient results for experiments: provide no experimental results of different Δ.**
>
> We agree with the reviewer, and thus conduct experiments on different Δ while keeping the same budget for selection process on SynLiDAR->POSS. The results are as follows
> Δ|0.05|0.1|0.15|0.2|0.25|0.3|0.35
> -|-|-|-|-|-|-|-
> AL|39.6|42.9|43.9|44.2|44.9|45.1|44.8
> ASFDA|40.0|46.2|46.0|48.0|48.2|48.3|48.0
>
> It can be seen that the performance of the large voxel grid is more robust to the noise and sparsity of point clouds.
>
> **Q6: Insufficient justifications: For example, about Voxel-centric selection, some justifications are missing in this paper. Why choosing voxel-based selection rather than other methods, e.g., point-based, range-image-based, BEV, etc (L101-102)? Any advantages and limitations of Voxel-based methods?**
>
> Thanks for the valuable question. At first, we would like to clarify that there exist image/frame-, region-, and point-based selection strategies in previous research. The first two require an offline stage, which may be infeasible at large scales. The last one is costly due to the sparsity of point clouds. By contrast, voxel-based selection aims to query the salient and exemplar regions and annotate all points within the region, which is more efficient and flexible. It can be easily applied to other methods, e.g., point-, voxel-, range-, BEV-view. Except for the results of voxel-view method reported in the main paper, we have added experimental results on a range-view method SalsaNext and a BEV-view method PolarNet in `Table R1 of the one-page PDF`. It is seen that our method still brings a large improvement using either range- or BEV-view method. Additionally, a drawback is that one may need to tune the voxel size for different scenarios. We find that the large voxel grid (Δ>0.2) is more robust to outdoor scenes in our experiments.
>
> **Q7: Minor issues: readability & variables definition.**
>
> Thank you. We will improve the readability of the paper and add a table with all the variables and their detailed descriptions.
>
> **Q8: Is the term 'cell' (L190) the same meaning as 'voxel' (L189)?**
>
> Yes, both "cell" and "voxel" denote the smallest unit of selection. We will use the same term "voxel" through the paper.

---

> > ### Comment · Reviewer_N28Y · 2023-08-14
> > **A good paper**
> >
> > Thank you for additional experiments. The authors have addressed all my concerns and questions.
> >
> > After reviewing the rebuttal, I am satisfied with the justifications provided for the voxel-centric online active learning approach. The additional ablation study on voxel size is useful for understanding the impact of the hyperparameter. I agree that the voxel-based selection strategy provides an efficient way to query salient regions in large 3D point clouds.
> >
> > I believe this is a solid contribution. The voxel-centric online active learning approach could be valuable for label-efficient LiDAR segmentation. Therefore, I am inclined to accept this paper and have raised my final rating.

---

> > > ### Author Response · Authors · 2023-08-16
> > > **Thanks for your help**
> > >
> > > We really appreciate your valuable suggestions and fruitful discussions. We hope this work can build a standard baseline and contribute to label-efficient LiDAR segmentation research.

---

### Official Review · Reviewer_zYAC · 2023-07-08

**Soundness:** 3 good
**Presentation:** 3 good
**Contribution:** 3 good
**Rating:** 7
**Confidence:** 4

**Summary:**

This paper proposes a voxel-centric online active learning baseline that efficiently reduces the labeling cost of enormous point clouds and effectively facilitates learning with a limited budget. The contribution of this paper can be summarized in three aspects:
1. A voxel-centric online active learning baseline that efficiently reduces the labeling cost of enormous point clouds and effectively facilitates learning with a limited budget.
2. A novel label acquisition strategy, voxel confusion degree (VCD), that requires 1000 times fewer annotations while reaching a close segmentation performance to that of the fully supervised counterpart.
3. A generally applicable and works for different network architectures (e.g., MinkNet, SPVCNN, etc.), in distribution or out of distribution setting (i.e., AL, ASFDA, and ADA), and simulation-to-real and real-to-real scenarios with consistent performance gains.

**Strengths:**

1. active learning is not entirely new in this field, but reducing the data volume requirement at the order of 1000 times is still very impressive
1. active learning coupled with domain adaptation has great practical significance, but little work has been done to consider the problem in 3D domains. The Annotator aims to minimize human labor in a new domain, regardless of whether samples from an auxiliary domain are available or not.
2. Methodologically, Annotator adopts a voxel-based representation for structured LiDAR data, which is different from the scan-based representation in UniDA3D.
3. Furthermore, Annotator can perform online active sampling that is more efficient than UniDA3D in terms of computation cost.

**Weaknesses:**

1. The cost of developing an AI based solution include multiple sectors: 1) data collection; 2) data annotation; 3) model training; 4) model deployment and integration. One of the most costly stage, both in time and money, is annotation. The AL paradigm, instead, proposes to use computation to reduce the total cost, by using computation to reduce the amount of data needed for human annotation. In this paper, an additional pre-training phase is added, it is unclear how to evaluate the cost balance, eg, increased cost of computation vs reduced cost of human annotation. In some sense, this is not the weakness/limitation of this particular paper, but rather applies to the whole AL paradigm.

**Questions:**

1. voxel based methods is one of the major and popular representations for lidar perception tasks, however there do exists other representations such as range view and point based methods, where the concept of voxel does not exist. question is, how does the proposed method apply to non-voxelization based methods?
2. segmentation tasks require dense point-wise annotation, but detection tasks only require, to some extent, much sparser box level annotations, how does the proposed method apply to detection based tasks? one would expect that the data volume reduction persists but not as impressive as 1000x

**Limitations:**

no potential negative societal impact is noticed by the reviewer to the best of their knowledge

---

> ### Author Rebuttal · Authors · 2023-08-09
>
> We sincerely thank the Reviewer zYAC for the detailed summary and constructive comments. We are glad that the reviewer acknowledges that the problem has great practical significance, the method is novel and generally applicable, and the experiments are very impressive. Here we answer all the questions and hope they can address the concerns.
>
> **Q1: In this paper, an additional pre-training phase is added, it is unclear how to evaluate the cost balance, eg, increased cost of computation vs reduced cost of human annotation. In some sense, this is not the weakness/limitation of this particular paper, but rather applies to the whole AL paradigm.**
>
> We agree with the reviewer that the trade-off between computation cost and annotation cost is a key problem in active learning. In this paper, we mainly focus on reducing the cost of human annotation while reaching a close segmentation performance to that of the fully supervised counterpart. Take the task of SynLiDAR $\to$ SemanticKITTI (KITTI) as an example, the simplest setup is performing active learning within KITTI dataset. In this case, no extra pre-training phase is included, but the performance is **less than satisfactory**. The reason might be the lack of prior information to select an initial annotated set (i.e. *cold-start problem*). To address this, we utilize some open-access datasets, especially synthetic dataset, to train an auxiliary model via a pre-training phase that serves as a warm-up stage, allowing for a smart selection of the data in the first round. Importantly the pre-training phase is trained with a few epochs and only conducted once, and the cost (both time and money) is **very low**. Here we provide a detailed analysis of the cost.
>
> phase|total epoch|running time (hours)|mIoU
> -|-|-|-
> pre-train on SynLiDAR|10|2.34|22.0
> AL on KITTI|50|18.04|53.7
> ASFDA on SynLiDAR $\to$ KITTI|50|18.39|54.1
> ADA on SynLiDAR $\to$ KITTI|50|28.48|**57.7**
>
> This demonstrates that our method can achieve a good balance between high performance and low cost (computation & annotation). In the future, we will explore more efficient ways to reduce the cost of both computation and annotation. We will add these results in the revision.
>
> **Q2: voxel based methods is one of the major and popular representations for lidar perception tasks, however there do exists other representations such as range view and point based methods, where the concept of voxel does not exist. question is, how does the proposed method apply to non-voxelization based methods?**
>
> Thanks for the valuable question. The main contribution consists of a plain voxel-centric online active learning baseline and a novel label acquisition strategy. As a general baseline, the proposed method can be easily applied to other non-voxelization based methods. Here, we have conducted experiments on a range-view method SalsaNext [a] and a BEV-view method PolarNet [b]. The per-class results on task of SynLiDAR$\to$POSS under AL setting with only 10 voxel budgets are shown in the following tables.
>
> SalsaNext:
> mehotd|pers.|rider|car|trunk|plants|traf.|pole|garb.|buil.|cone|fence|bike|grou.|mIoU
> -|-|-|-|-|-|-|-|-|-|-|-|-|-|-
> Random|22.8|10.4|30.9|15.6|66.6|8.3|5.5|0.0|57.2|10.6|26.6|40.6|74.7|28.4
> Entropy|33.1|18.6|32.8|20.1|64.7|11.2|5.5|12.7|52.6|4.3|40.2|45.2|76.8|32.1
> Margin|33.6|28.0|29.8|24.5|61.3|20.6|12.5|18.4|48.0|0.0|27.7|38.2|71.1|31.8
> **Ours**|39.7|31.8|32.0|26.2|64.7|17.7|11.8|13.7|53.9|13.1|40.9|45.1|76.5|**35.9**
> Target-only|52.7|40.2|39.2|28.1|71.5|28.3|18.7|8.0|66.1|16.7|50.1|51.0|79.3|42.3
>
> PolarNet:
> mehotd|pers.|rider|car|trunk|plants|traf.|pole|garb.|buil.|cone|fence|bike|grou.|mIoU
> -|-|-|-|-|-|-|-|-|-|-|-|-|-|-
> Random|40.8|0.1|36.6|3.1|74.0|1.3|17.8|0.0|69.3|0.0|50.3|50.5|76.1|32.3
> Entropy|50.3|11.8|44.5|10.0|71.0|19.1|13.3|0.0|63.6|0.0|45.4|48.8|77.9|35.0
> Margin|42.8|24.3|22.0|19.4|64.2|17.5|20.1|4.0|54.7|0.0|33.0|35.4|64.1|30.9
> **Ours**|55.9|39.2|44.4|22.4|70.3|28.7|18.6|6.9|64.3|21.7|51.9|51.7|76.2|**42.5**
> Target-only|62.3|51.8|66.3|22.8|75.5|29.4|21.8|4.8|74.9|46.1|61.3|57.2|80.8|50.4
>
> It is seen that our method still brings a large improvement using either range- or BEV-view backbones with only the limited budget. However, **the performance gains are less than that of the voxel-view counterparts**. Also, to achieve 85% fully-supervised performance, the budget are twice. We think that some annotations of voxel-centric selection might be invalid in other non-voxelization representations. We will add these discussions in the revision for better understanding.
>
> [a] Cortinhal et al. Salsanext: Fast, uncertainty-aware semantic segmentation of lidar point clouds. In ISVC 2020.
>
> [b] Zhang et al. Polarnet: An improved grid representation for online lidar point clouds semantic segmentation. In CVPR 2020.
>
> **Q3: segmentation tasks require dense point-wise annotation, but detection tasks only require, to some extent, much sparser box level annotations, how does the proposed method apply to detection based tasks? one would expect that the data volume reduction persists but not as impressive as 1000x.**
>
> This question is very interesting and valuable. For active learning in 3D object detection task, most strategies are offline, i.e., a few frames are first selected from the entire dataset, and then all boxes in each frame are annotated manually. Our Annotator, on the other hand, is an online strategy that allows querying and annotating within each frame.
>
> Therefore, some changes need to be made to adapt to the detection task. 1) frame-level selection should be used; 2) VCD should be reformulated as follows: in each frame, obtain pseudo-label for each box, collect statistical information on the predicted categories of all boxes, and VCD is computed based on the percentage of boxes belonging to each different category. In other words, in the original algorithm, a frame is a voxel and a box is a point of the voxel. We will explore the application to other LiDAR perception tasks in the future.

---

### Author Rebuttal · Authors · 2023-08-10

Dear reviewers and AC,

We sincerely thank all the reviewers for their positive comments and helpful feedback that have certainly helped improve the quality of this paper. We have uploaded the responses w.r.t. each reviewer together with `the one-page PDF`.

In response to the comments, we have carefully revised and enhanced the manuscript with the following additional discussions and experiments:


1. Add discussion about the cost (computation v.s. annotation) balance in AL paradigm.
2. Provide additional experiments that apply the proposed method to both range-view and BEV-view methods.
3. Provide additional comparisons with existing methods including two AL methods for image classification, two AL methods for outdoor LiDAR semantic segmentation and one for indoor scenes.
4. Add the error bars for the results.
5. Add additional experiments to analyze the effect of voxel size.
6. Provide the analysis of training the network with only a handful of points.
7. We shed light on some potential applications of our Annotator, such as indoor semantic segmentation, LiDAR object detection task.

We hope our response sincerely address all the reviewers' concerns.

Thank you very muck for your time and consideration.

Best regards,

Submission2070 Authors.

---

### Decision · Program_Chairs · 2023-09-21

**Decision:**

Accept (poster)

**Comment:**

This paper introduces intriguing work for active learning for 3D semantic segmentation. Reviewers favor the proposed approach and the demonstrated results (especially the results on the cross-domain experiments). The authors provided comprehensive feedback regarding specific questions, and the authors provided a supplement pdf file. In particular, there were concerns from reviewer KrDz, but AC notes that the authors adequately defend it. After the rebuttal phase, all reviewers voted for the high scores, and AC saw that there were constructive suggestions for the paper. AC recommends paper acceptance. Applying the valuable discussion and additional material in the revision is strongly advised.